# SuRe: Surprise-Driven Prioritised Replay for Continual LLM Learning

## Abstract

Continual learning, one's ability to adapt to a sequence of tasks without forgetting previously acquired knowledge, remains a major challenge in machine learning and a key gap between artificial and human intelligence. While regularisation and replay perform well in vision, they lag behind multi-task learning for large language models (LLMs), especially at scale with many tasks. We revisit replay and argue that two failure modes drive this gap: selection (what to rehearse) and integration (how to consolidate new knowledge). To address selection, we propose Surprise-prioritised Replay (SuRe), a simple, architecture-agnostic rule that ranks and stores the most surprising (high Negative Log-Likelihood) sequences. SuRe achieves state-of-the-art performance in the Large Number of Tasks (LNT) setting and delivers the best overall average across both Standard CL and LNT benchmarks. To address integration, we add a dual-learner design with fast and slow LoRA adapters merged via an exponential moving average (EMA), enabling rapid adaptation while stabilising long-term knowledge. Combining SuRe with the dual learner yields further gains, including improvements of up to +5 accuracy points on LNT over prior SOTA. Ablation studies confirm that our proposed method remains robust under reduced replay frequency and small buffer size, demonstrating both effectiveness and sample efficiency. Taken together, our results establish replay as a strong baseline for continual LLM fine-tuning and demonstrate that surprise-based selection and slow-weight consolidation are complementary components for mitigating catastrophic forgetting.

## 1 Introduction

By nature, humans can easily learn new information and acquire new skills one at a time with few examples, an ability which is often taken for granted but proves extremely difficult for machine learning models. This problem framing, often referred to as continual or lifelong learning (CL), has attracted increasing attention as model capabilities have advanced over the past decade. Early research of CL in deep learning focused primarily on Vision and Reinforcement Learning (RL) tasks (Kirkpatrick et al., 2017; Aljundi et al., 2018; Rolnick et al., 2019). Recently, the field has expanded toward Natural Language Processing (NLP), motivated by the rapid rise of large language models (LLMs).

While most machine learning models are static by design, a recent shift in paradigm, thanks to advances in In-Context Learning, has shown a promising avenue for more adaptable models (Sun et al., 2025; Zhang et al., 2025; Yang et al., 2025b). While allowing for more flexible models that can leverage current context to NLP tasks, these advances remain limited by the effective size of the context window. CL goes further when it comes to designing flexible models, as it not only requires effective adaptation to new datasets (plasticity) but also effective retention of previously acquired skills (stability). This plasticity-stability dilemma (Mermillod et al., 2013) is at the centre of CL, with one of the main challenges being a lack of stability, leading to catastrophic forgetting (McCloskey & Cohen, 1989; Ratcliff, 1990), performance on previously trained datasets drops as new tasks, domains or classes are introduced. These three framings, referred to as *Task-Incremental*, *Domain-Incremental* and *Class-Incremental* respectively (van de Ven et al., 2022), each come with their own challenge. Often, because the label space expands over time while task identity is unavailable at test time, forcing the model to distinguish among old and new classes without seeing them jointly, class-incremental is considered to be the hardest setting (van de Ven & Tolias, 2019). This is especially the case when

dealing with large number of tasks, a setting where most methods rely on known task identity during training, but still struggle to match the performance of Multi-Task Learning (MTL).

Here, we formalise catastrophic forgetting as the sum of two complementary sources of error: (i) selection error from imperfect replay distribution estimates, and (ii) integration error from how new knowledge updates are consolidated. We show these errors are additive and complementary, the strongest methods against catastrophic forgetting should address both.

To explore this idea in practice, we first focus on replay, one of the simplest solutions to the selection problem. We show that previous studies of LLM CL often underestimate replay's performance and effectiveness in the *Class-Incremental* scenario. In particular, prior comparisons often mix online, task-agnostic replay with methods that assume known task boundaries (Wang et al., 2023; Qiao & Mahdavi, 2024), potentially leading to lower performance and unfair comparison. We therefore evaluate replay under the same assumption set (known boundaries), yielding a fairer comparison. Notably, we find that under fair comparison, surprise-based selection outperforms random replay and achieves state-of-the-art results in the LNT setting, while also providing the strongest overall average performance across both Standard CL and LNT benchmarks. Finally, for integration error, we show that a simple exponential moving average (EMA) approach, which stabilises the consolidation of new representations, yields complementary improvements. This confirms our additive error hypothesis, and combining SuRe with EMA further improves performance, particularly in the LNT setting, achieving gains of up to +5 points over prior state-of-the-art.

Our contributions can be summarised as follows: (1) We formalise forgetting as the sum of selection and integration errors, motivating complementary mechanisms for each. (2) We propose Surprise prioritised Replay (SuRe) to improve sample selection efficiency. (3) We show that combining SuRe with a simple integration mechanism (EMA), following the dual-learning framework (Pham et al., 2021; Gao et al., 2023), yields strong overall performance, achieving SOTA in LNT and the best average across both benchmarks, empirically confirming (1).

## 2 RELATED WORK

Three lines of research have emerged to approach catastrophic forgetting, replay, regularisation and architecture. These were first developed in the vision and Reinforcement Learning literature before being adapted to the modern architecture of Large Language Models (LLMs) and Vision Language Models (VLMs). For the purpose of efficiency, we focus on replay and methods which were introduced in the CL with LLM literature.

**Replay Based Methods.** Each replay method can be described by a few design choices: how is the buffer updated, which samples should be replayed and when, and, are the rehearsed samples stored or generated. Experience Replay (ER) (Rolnick et al., 2019; Chaudhry et al., 2019) is the simplest and often the most effective approach. The buffer is updated via reservoir sampling (Vitter, 1985) so that each incoming raw example has equal probability of being stored, and sequences are then drawn uniformly at random. Many subsequent methods can be viewed as variants of ER. Isele & Cosgun (2018) focused on Reinforcement Learning and compared different update rules, including keeping the most surprising traces or those leading to the highest rewards. Their experiments showed that reservoir performed best, which is aligned with Araujo et al. (2022) who repeated this comparison in a CL with LLMs setting with surprise selection performing poorly in both instances. InfoRS (Sun et al., 2022) went a step further by introducing a information theory based update rule. They combined two rules, effectively keeping the most surprising and learnable samples. Here, a sample is regarded as surprising with regard to the content of the current memory module, and is computed as the posterior from a small Bayesian linear model. On the other hand, the learnability criteria is used to discard outliers by computing how well the updated model predicts the point's own label and discarding poor predictions. Tackling another design choice, Maximally Interfered Retrieval (Aljundi et al., 2019) keeps the update rule as a Reservoir and focuses instead on which samples to replay. For each batch, they estimate which samples in the memory would face the highest increase in loss if the model were to be trained on it. Replaying these selected samples thus acts as a stronger regulariser for that specific gradient step. Finally, Generative Replay (Shin et al., 2017) and LAMOL (Sun et al., 2019) both proposed approaches which generate samples from past distributions to avoid storing raw

samples. While the methods that focused on improving the update and sampling rules are online and did not require task identity during training, LAMOL requires it during training.

**Parameter-Efficient Continual Learning.** Given the compute and memory cost of fine-tuning billion-parameter models, many NLP CL methods adopt parameter-efficient fine-tuning (PEFT). The most common approach being LoRA (Hu et al., 2021), which adds trainable low-rank adapters to attention/feedforward projections to approach full fine-tuning performance while updating only a small fraction of parameters. This parameter efficient fine-tuning solution has thus been used as the basis of many CL methods in the NLP literature. This is the case of O-LoRA (Wang et al., 2023) which introduces new LoRA heads for each dataset and adds a penalty which guarantees orthogonal solutions for each LoRA pairs. Learn More but Bother Less (Qiao & Mahdavi, 2024) takes this idea further by initialising each PEFT module based on previous task, facilitating forward transfer and avoiding interference. Taking a different approach to the problem of parameter efficient CL, Progressive Prompts takes inspiration from prompt tuning models and learns a new prompt embedding per task while leaving the base weights frozen. Lastly, recent works have investigated how model merging could be used in the context of CL. This led to Hickok (2025) proposing to use Exponential Moving Average (EMA) (Tarvainen & Valpola, 2017), along with other sequential merging approaches, as ways to smoothly regularise the LLMs' learning process.

## 3 METHODS

### 3.1 SELECTION–INTEGRATION DECOMPOSITION

We formalise forgetting as the sum of a *selection mismatch* term (how well replay approximates the past distribution) and an *integration* term (variance/instability in how new updates are consolidated). We work in the LoRA subspace under standard local assumptions (smoothness and PL near the trajectory). Full proofs are deferred to Appendix §H.

**Setup and notation.** Tasks arrive $1, \ldots, t$. Let $R_k(\theta) = \mathbb{E}_{z \sim P_k} \ell(\theta; z)$ (where $z$ is a single example and $\ell$ the per-example loss, e.g., cross-entropy/sequence NLL), and let $P_{1:t-1} = \frac{1}{t-1} \sum_{k<t} P_k$ be the uniform mixture of past tasks. A replay buffer induces a distribution $q$ with replay risk $\tilde{R}_{1:t-1}(\theta) = \mathbb{E}_{z \sim q} \ell(\theta; z)$. Let $\mathcal{F}_{\text{loc}} = \{\ell(\theta; \cdot) : \theta \text{ in a local neighbourhood}\}$ and let $D_{\mathcal{F}_{\text{loc}}}$ be any integral probability metric (IPM) over $\mathcal{F}_{\text{loc}}$ (e.g., MMD). The slow model is a *consolidated* iterate obtained by a stable averaging operator $\mathcal{A}_\psi$ over fast iterates, e.g., an exponential moving average (EMA) with rate $\beta \in (0, 1)$. *Forgetting* $\mathcal{F}$ denotes any standard average forgetting metric (e.g., AP−FP or Chaudhry AF); our bound applies to such monotone summaries.

**Lemma 1 (Selection mismatch via IPM)** *For all $\theta$ in the local region,*

$$\left| \tilde{R}_{1:t-1}(\theta) - R_{1:t-1}(\theta) \right| \leq D_{\mathcal{F}_{\text{loc}}}\big(P_{1:t-1}, q\big). \tag{1}$$

**Lemma 2 (EMA reduces integration variance)** *Let $\theta_{\text{fast}}^{(n)}$ be SGD iterates on the mixed objective and $\theta_{\text{slow}}^{(n)} = \beta \, \theta_{\text{slow}}^{(n-1)} + (1 - \beta) \, \theta_{\text{fast}}^{(n)}$. Under local L-smoothness and a $\mu$-PL condition, there exist constants $C_b, C_v, C_d > 0$ such that for any past task $k < t$,*

$$\mathbb{E}\big[R_k(\theta_{\text{slow}}^{(n)}) - R_k(\theta_k^\star)\big] \leq C_b(1 - \beta) + C_v \frac{1}{(1 - \beta)} \frac{\sigma^2}{\mu n} + C_d \, \delta, \tag{2}$$

*where $\sigma^2$ is the SGD noise level and $\delta$ bounds drift between task optima.*

Remark: Comparing the bounds for $\beta = 0$ (Single Learner) versus $\beta \to 1$ (Slow Learner), we see that a single learner has the variance term proportional to the raw SGD noise $\sigma^2$. In contrast, the Slow Learner scales the variance term by $(1 - \beta)$, which, for $\beta = 0.995$, reduces the effective variance contribution to the loss bound by a factor of approximately 200. This illustrates the advantage of the slow averaging mechanism in controlling integration variance.

**Theorem 1 (Additive bound; complementary controls)** *Summing effects across tasks, the expected forgetting of the slow model satisfies, in a local region,*

$$\mathbb{E}\,\mathcal{F} \;\leq\; \underbrace{A \cdot D_{\mathcal{F}_{\text{loc}}}\big(P_{1:T-1}, q\big)}_{\text{selection (replay) term}} + \underbrace{B(\psi) \cdot \frac{\sigma^2}{\mu N}}_{\text{integration (consolidation) term}} + \underbrace{C\,\Delta_{\text{drift}}}_{\text{nonstationarity}}, \tag{3}$$

*for constants $A, B, C > 0$ and total fast steps $N$. With finite memory and finite $N$, neither addend can be driven to zero by tuning the other alone; thus replay (selection) and EMA (integration) provide complementary benefits to continual learning.*

*Remark.* Appendix §H proves Lemma 2 for EMA; other consolidation operators $\mathcal{A}_\psi$ fit the same bound by replacing the mechanism-specific factor $B(\psi)$ accordingly (for EMA, $B(\psi) \equiv B(\beta)$; related examples include Polyak averaging, SWA, and model soups).

**Selection Error** ($A \cdot D_{F_{loc}}$)**:** This term quantifies the mismatch between the replay buffer distribution $q$ and the true past task distribution $P_{1:T-1}$. Uniform sampling (reservoir) treats all past samples as equally important for representing the loss landscape. In high-dimensional LLMs, this is inefficient—most samples lie in flat, well-learned regions with low gradient norms.

**Integration Error** ($B(\psi) \cdot \sigma^2/(\mu N)$)**:** This term captures the instability introduced by stochastic gradient noise when learning new tasks. The fast learner performs SGD on small batches with noise variance $\sigma^2$. This high-variance trajectory leads to "plasticity" that overwrites old knowledge—the core of catastrophic forgetting. $B(\psi)$ quantifies the variance-reduction factor of the consolidation operator.

As a design implication of the above, any buffer policy that lowers $D_{\mathcal{F}_{\text{loc}}}(P, q)$ tightens the selection term; any consolidation that lowers the variance factor $B(\psi)$ tightens the integration term (for EMA, $B(\psi) \equiv B(\beta)$). In §3.2–§3.3 we instantiate these with a simple surprise-based replay policy and EMA dual adapters.

### 3.2 SURPRISE REPLAY

Following these theoretical motivations, we introduce a new update rule for replay algorithms and combine it with a dual set of learners which are merged via EMA. The overall proposed method is shown in Figure 1 and the pseudocode is provided in Algorithm 1.

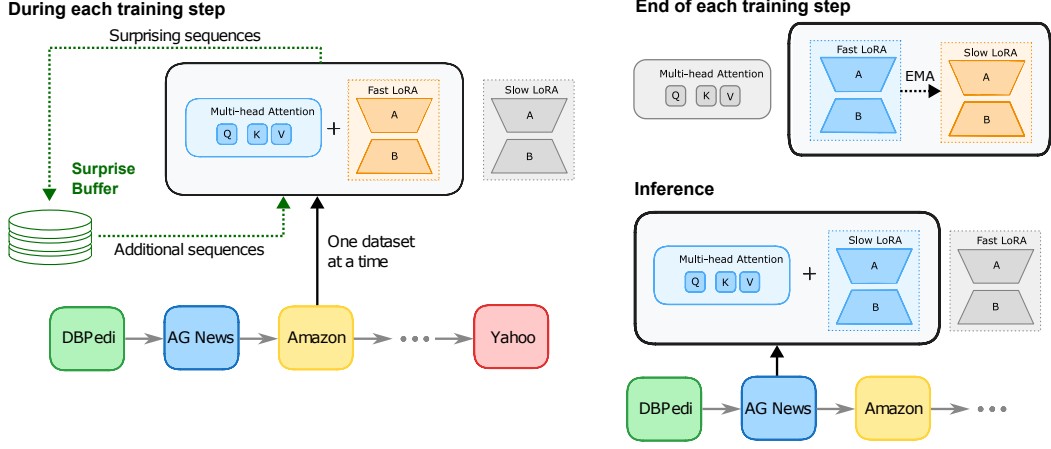

Figure 1: 1. During training the base and slow LoRA weights are frozen, while the fast LoRA is updated on current samples plus replayed examples from the surprise buffer. The buffer is updated to retain the most surprising samples per task. 2. After each step, the fast and slow LoRA weights are merged via an exponential moving average. 3. At inference, only the base model and the slow learner are used for prediction.

Currently, the standard approach in the literature for a replay algorithm is reservoir sampling (Vitter, 1985; Rolnick et al., 2019; Chaudhry et al., 2019), which maintains a representative subset of samples by giving each sample equal probability of being retained. These simple yet effective methods have been shown to be the most appropriate in many cases, often outperforming more complex alternatives.

However, memory consolidation in the brain is known to be non-uniform, and replay to be selective. In particular, surprise has been shown to be a key driver of memory retention and replay (Momennejad et al., 2018; Jang et al., 2019; Lindsey & Litwin-Kumar, 2024). Such findings suggest that surprising events are likely harder to learn and more susceptible to forgetting, and thus are more valuable for selective memory consolidation. Similar intuitions have proven effective in reinforcement learning through prioritised replay (Peng & Williams, 1993; Moore & Atkeson, 1993; Schaul et al., 2016) and surprise-based episodic memory (Zakharov et al., 2021; Coda-Forno et al., 2024), as well as in organising episodic memory structures in LLMs (Fountas et al., 2025; Behrouz et al., 2024).

Building on these insights, we hypothesise that replaying the most surprising sequences when training a model in CL settings provides three benefits. First, it directs computation on the sequences that lead to large prediction errors, occur infrequently (and thus are underrepresented), or sit at task boundaries where interference is highest. This gives the model more chances to properly learn these high-loss, easy-to-forget examples. Second, it preserves efficiency by enabling lower replay frequencies without sacrificing performance, since the retained samples act as a compact but representative regulariser of past tasks. Finally, by prioritising high-NLL samples, SuRe performs implicit importance sampling. High-NLL sequences have large $\|\nabla\ell(\theta; z)\|$, meaning they contribute disproportionately to the true gradient $E_P[\nabla\ell]$. Storing these samples ensures the buffer approximates the gradient geometry of past tasks, not just the data frequency. This directly reduces $D_{F_{loc}}(P, q)$, tightening the selection term in Equation 3.

Thus, we propose to replace uniform buffer updates with surprise-based replay, where storage decisions are guided by the Bayesian surprise of each input sequence. For a given input $x_i$ with tokenised sequence $z_i = (z_{i,1}, ..., z_{i,T_i})$, surprise is measured as the negative log-likelihood under the model:

$$s_\theta(z_i) = -\frac{1}{T} \sum_{t=1}^{T_i} \log p_\theta(z_{i,t} \mid z_{i<t}, x_i) \tag{4}$$

Following Rolnick et al. (2019) we set our buffer size to 2% of the overall dataset size. We allocate an equal per-task quota which depends on the number of tasks currently in the buffer, irrespective of dataset size per task. Thus, after training on d datasets, each task has $m_i^{(d)} = [S/d]$ samples in the buffer, with S as our buffer size. In practice, surprise-based replay is architecture-agnostic and can be applied in any continual learning setting or modality (e.g., vision, speech, video, text). In the context of LLMs, it is particularly effective when combined with parameter-efficient fine-tuning methods such as LoRA, further reducing computational cost.

### 3.3 DUAL LEARNERS

Replay can easily be combined with other approaches. Here we decided to focus on a dual learner architecture (Pham et al., 2021; Ran et al., 2025) with exponential moving average in line with Gao et al. (2023); Hickok (2025). At the start of training we freeze the base model and, for each attention layer's $W_Q$ and $W_V$, attach two LoRA adapters: a fast head and a slow head. Each head is a low-rank pair $(A, B)$, we initialise $A$ randomly and set $B = 0$ following Hu et al. (2021). Before training on dataset $D_i$, we compute the surprise (Eq. 4) for each sequence $x \in D_i$ and insert the $m_i$ most surprising sequences into a replay buffer $M$. The fast adapters are then updated by minimising the cross-entropy on the union batch $B_t \subset D_i \cup M$ while the slow adapters are not directly optimised but updated via an exponential moving average (EMA),

$$\theta_t^{\text{slow}} \leftarrow \beta\theta_{t-1}^{\text{slow}} + (1 - \beta)\theta_t^{\text{fast}}, \qquad \beta \in (0, 1). \tag{5}$$

Equation 5 can be rewritten as $\theta_t^{slow} = (1 - \beta) \sum_{k=0}^{t} \beta^k \theta_{t-k}^{fast}$. That is, the slow parameters are a geometrically weighted ensemble of recent fast iterates with effective window length $\approx 1/(1 - \beta)$ Equivalently, $\theta_t^{slow}$ is the unique minimiser of the exponentially weighted least-squares fit to the

history of fast parameters:

$$\theta_t^{slow} = arg\min_\theta \sum_{k=0}^{t} \beta^{t-k}\|\theta - \theta_k^{fast}\|_2^2 \tag{6}$$

so the slow learner implements a low-pass filter on parameter trajectories, reducing iterate variance while introducing a controllable tracking lag. In the non-stationary setting introduced by task sequence $D_i$ and replay $M$, this two timescale design lets the fast adapters adapt rapidly to $D_i$, while the slow adapters aggregate only changes that persist across many steps, thereby mitigating catastrophic forgetting. In our bound, $B(\beta) = \frac{1}{1-\beta}$ appears as a coefficient, larger $\beta$ (e.g., 0.995) means stronger averaging, reducing the effective noise and tightening the integration term in Equation 3.

---

**Algorithm 1** Dual-LoRA with Surprise Replay

---

1: Input data stream $\mathcal{D}$, memory $\mathcal{B}$ (cap $B_{max}$), replay interval $k$, EMA rate $\beta$
2: Initialise $\theta^{fast}$, $\theta^{slow}$ (LoRA on Q,V, random init)
3: **for** $t \in \{1,\ldots,T\}$ **do**
4:    Select top-$p$ surprising samples $\mathcal{C}$ from $\mathcal{D}_t$ (NLL under $\theta^{fast}$)
5:    **for** $s \in$ training steps on $\mathcal{D}_t$ **do**
6:        Sample batch $\mathcal{B}_{cur} \subset \mathcal{D}_t$ $\hspace{3cm}$ ($|\mathcal{B}_{cur}| = 64$)
7:        **if** $s \mod k = 0$ **then**
8:            Sample $\mathcal{B}_{rep} \subset \mathcal{B}$ $\hspace{3.5cm}$ ($|\mathcal{B}_{rep}| = 32$)
9:            $\mathcal{B}_{mix} \leftarrow \mathcal{B}_{cur} \cup \mathcal{B}_{rep}$
10:       **else**
11:           $\mathcal{B}_{mix} \leftarrow \mathcal{B}_{cur}$
12:       **end if**
13:       $\mathcal{L}_{CE} \leftarrow$ cross-entropy$(\mathcal{M} \oplus \theta^{fast}, \mathcal{B}_{mix})$
14:       $\theta^{fast} \leftarrow \theta^{fast} - \eta\nabla\mathcal{L}_{CE}$
15:       $\theta^{slow} \leftarrow \beta\theta^{slow} + (1-\beta)\theta^{fast}$
16:    **end for**
17:    $\mathcal{B} \leftarrow$ UpdateBuffer$(\mathcal{B}, \mathcal{C}, B_{max})$
18: **end for**

---

## 4 EXPERIMENTS

### 4.1 EXPERIMENTAL SETUP

**Standard CL Benchmark.** We first evaluate our methods on the Standard CL Benchmark, one of the most widely used setups for CL evaluation of LLMs (Qin & Joty, 2022). This benchmark is composed of four text classification datasets: AG News, Amazon Reviews, DBpedia, and Yahoo Answers, which were all introduced by Zhang et al. (2016). Following (Wang et al., 2023), we randomly select 5,000 training samples and 500 test sequences.

**Large Number of Tasks.** To assess performance in a more challenging and long term scenario, we further extend the benchmark by adding 11 additional datasets, following Razdaibiedina et al. (2023); Wang et al. (2023); Qiao & Mahdavi (2024). These datasets are: MNLI, QQP, RTE and SST-2 from Wang et al. (2019) as well as WiC, CB, COPA, BoolQ, MultiRC and IMDB from Wang et al. (2020). In line with prior work (Razdaibiedina et al., 2023; Wang et al., 2023; Qiao & Mahdavi, 2024) we randomly sample 1,000 training examples and 500 test samples from each dataset.

**Metrics.** In order to measure and compare the performance of each approach, we report the final performance. This measures the average performance across all tasks after training on the final task. It primarily reflects stability, that is how well the model retains knowledge at the end of training. Let $N$ be the total numbers of tasks, and let $a_{T_j}^N$ denote the accuracy on task $T_j$ evaluated after training the model on task $N$. Then: $FP = \frac{1}{N}\sum_{j=1}^{N} a_{T_j}^N$

**Baselines.** For an effective evaluation of the introduced methods we picked a number of baselines which are either regarded as standard approaches in the field or are recent state-of-the-art approaches on either of the benchmarks we are evaluating our method against.

- Multi-Task Learning (MTL) (Caruana, 1997): jointly trains the model on all datasets at once, allowing optimal sharing of representations and transfer across tasks. This is not per se a CL method but it is often regarded as the upper bound of CL performance.

- EWC (Kirkpatrick et al., 2017): estimates parameters importance using the Fisher Information Matrix and constrains important parameters from drifting too far when learning new tasks.

- Experience Replay (Rolnick et al., 2019; Chaudhry et al., 2019): retains 2% samples from each datasets which are then replayed when fine-tuning the model on a new training set.

- Leitner Replay (M'hamdi & May, 2024): selects replay samples using a dynamic Leitner-style skill rating system that prioritises examples based on how well the model has learned them.

- AimMerging (Feng et al., 2025): adaptively merges intermediate models by tracking parameter-change signals and replay-based forgetting signals, using stored past data to trigger merges when historical loss rises.

- O-LoRA Wang et al. (2023): introduce a new set of LoRAs for each dataset, these adapters are then trained on the current dataset with an orthogonal constraint before being merged into the main model.

- LB-CL(Qiao & Mahdavi, 2024): trains low rank adapters using singular value decomposition. The low rank parameters are initialised using a sensitivity score enabling forward transfer. Additionally, they project gradients from the new tasks into orthogonal subspaces to avoid interference.

- N-LoRA (Yang et al., 2024): encourages extremely sparse, non-colliding low-rank updates so each task occupies its own parameter subspace, reducing interference during continual learning.

- O-LieRA(Cao & Wu, 2025): applies orthogonal low-rank updates within a Lie-group multiplicative framework to preserve parameter geometry while preventing task-to-task interference.

- Mixture-of-Rank Adaptation (MoRA) (Lu et al., 2025): decomposes low-rank adapter into rank-1 components and treats them as independent experts. A self-activated sparse gating mechanism then selects only a small, input dependent subset of these ranks during training and inference.

- Progressive Prompts (Razdaibiedina et al., 2023): learns a soft-prompt per task instead of finetuning LoRA parameters, updating fewer than 0.1% of model parameters.

## 4.2 MAIN RESULTS

The main results of our experiments are summarised in Table 1. First, we find that replay based methods perform better than is often acknowledged in the literature. Even with a simple reservoir sampling strategy, and having a 1:2 replay ratio, this method achieves competitive performance across both benchmarks, consistently outperforming regularisation based methods such as EWC and O-LoRA. This suggests that replay remains a highly effective and reliable baseline for CL in LLMs, and requires further investigation. Secondly, we can observe that our proposed Surprise Replay strategy consistently improved the rehearsal performance. The benefits are particularly strong in the LNT setting, where task diversity and limited per-task data accentuate catastrophic forgetting. Here, Surprise Replay significantly improves over uniform replay, highlighting the relevance of selective memory updating in more realistic CL scenarios. Third, our results show that the Dual Learner architecture further enhances performance. While a dual learner with vanilla replay is already competitive, the Surprise Dual Learner improves by over 5 percentage points compared to the previous state of the art method. This clearly narrows down the gap to the Multi-Task Learning upper bound and motivates further work in that direction.

To further compare our proposed methods with recent approaches, we followed the hyperparameter settings reported in Cao & Wu (2025) and present the results in Table 11. Since no batch size was specified, we follow common practice, report results using a batch size of 64 in Table 11 and include a more complete table in the in the Appendix B.1. Our methods consistently outperform prior approaches across all batch configurations, demonstrating robustness of our selection–integration approach. Finally, looking at measures of forgetting (Table 9), we find that Slow Replay maintains strong stability with negative forgetting, while the surprise replay was already facing reduced forgetting compared to it's random counterpart.

| | **Standard CL Benchmark** | | | | **Large Number of Tasks** | | | |
| --- | --- | --- | --- | --- | --- | --- | --- | --- |
| | Order-1 | Order-2 | Order-3 | avg | Order-4 | Order-5 | Order-6 | avg |
| SeqFT$^\diamond$ | 18.9 | 24.9 | 41.7 | 28.5 | 7.4 | 7.3 | 7.4 | 7.4 |
| SeqLoRA$^\diamond$ | 39.5 | 31.9 | 46.6 | 39.3 | 4.9 | 3.5 | 4.2 | 4.2 |
| EWC$^\diamond$ | 46.3 | 45.3 | 52.1 | 47.9 | 44.9 | 44.0 | 45.4 | 44.8 |
| O-LoRA$^\diamond$ | 74.9 | 75.3 | 75.9 | 75.4 | 70.0 | 65.5 | 70.5 | 68.8 |
| LB-CL$^\diamond$ | 76.9 | 76.5 | 76.8 | 76.7 | 68.4 | 67.3 | 71.8 | 69.2 |
| Reservoir Replay | 76.8 | 77.7 | 76.5 | 76.9 | 69.6 | 69.4 | 68.3 | 69.1 |
| MoRA$^\spadesuit$ | 77.4 | 77.5 | **77.9** | 77.6 | 68.9 | 68.3 | 72.0* | 69.7 |
| Surprise Replay | 77.0 | 78.1* | 76.4 | 77.2 | 72.8 | 72.1* | 71.6 | 72.1 |
| Slow Reservoir Replay | 78.0* | 78.0 | 77.3* | 77.8* | 74.0* | 71.9 | 71.6 | 72.5* |
| Slow Surprise Replay | **78.8** | **78.5** | 76.9 | **78.1** | **75.6** | **74.8** | **75.0** | **75.1** |
| ProgPrompt$^\diamond$ | 76.1 | 76.0 | 76.3 | 76.1 | 78.7 | 78.8 | 77.8 | 78.4 |
| PerTaskFT$^\diamond$ | 70.0 | 70.0 | 70.0 | 70.0 | 78.1 | 78.1 | 78.1 | 78.1 |
| MTL$^\diamond$ | 80.0 | 80.0 | 80.0 | 80.0 | 76.3 | 76.3 | 76.3 | 76.3 |

Table 1: Final accuracy (%) on the Standard CL and the Large Number of Tasks Benchmarks for different baselines on T5. $^\diamond$ and $^\spadesuit$ indicate results taken from Qiao & Mahdavi (2024) and Lu et al. (2025) respectively **Bold** indicates the best results and * is for the second best.

| | **Standard CL Benchmark** | | | | **Large Number of Tasks** | | | | **All** |
| --- | --- | --- | --- | --- | --- | --- | --- | --- | --- |
| | Order-1 | Order-2 | Order-3 | avg | Order-4 | Order-5 | Order-6 | avg | avg |
| N-LoRA$^\diamond$ | 79.2* | 78.4* | 78.8* | 78.8* | 73.6 | 70.3 | 73.2 | 72.4 | 75.6 |
| OLieRA$^\diamond$ | **79.9** | **79.5** | **79.5** | **79.6** | 73.8 | 70.4 | 73.5 | 72.6 | 76.1 |
| AimMerging | 71.2 | 72.4 | 70.9 | 71.5 | 74.1 | 73.5 | 73.7 | 73.7 | 72.6 |
| Leitner Replay | 74.0 | 73.8 | 72.0 | 73.3 | 75.1 | 74.8 | 76.9 | 75.6 | 74.5 |
| Surprise Replay | 78.4 | 77.0 | 75.8 | 77.1 | 77.6* | 76.2* | 78.0* | 77.3* | 77.2 |
| Slow Surprise Replay | 78.8 | 77.9 | 77.6 | 78.1 | **78.0** | **77.0** | **78.8** | **77.9** | **78.0** |

Table 2: Final accuracy (%) on the Standard CL and the Large Number of Tasks Benchmarks for different baselines on T5. Here the hyperparameters used are the ones reported by Cao & Wu (2025). $^\diamond$ indicate the results were taken from Cao & Wu (2025).

## 4.3 ABLATION STUDIES

Having established that Surprise Replay and the Surprise Dual Learner achieve state-of-the-art performance compared to strong baselines, we next examine which design choice drive these improvements. We compare computing surprise on labels versus full sequences, analyse the effect of when surprise is computed and when the buffer is updated, explore dynamic updates of surprise values during replay and finally benchmark our approach against classical replay methods such as Reservoir and Gradient-Based Sample Selection. Our results are summarised in Table 3.

**Surprise on Labels vs. Full Sequences.** Results show that label level surprise performs poorly on both benchmarks (64.9% on the Standard CL Benchmark and 61.2% on the LNT setting), indicating that this signal is too weak to guide selective replay effectively. As labels are only one or a few words, it is most likely that only the most surprising classes will be kept in the buffer leading to a massive imbalance when later replaying some sequences. On the other hand, some classes, not necessarily well classified ones, might never enter the buffer and will largely degrade performance on downstream task.

**When to Compute Surprise and When to Update the Buffer.** Here, our results suggest that the choice of when to update the buffer had a stronger effect than the timing of surprise computation. Indeed, both Surprise Before-Update After and Surprise After achieved significant gains compared to performing both actions before training on each dataset. Adding samples to the buffer after training potentially improves the regularisation by ensuring that previous tasks are replayed more often. On the other side, updating the buffer before training will lead the model to focus more on the current task while decreasing the amount of natural regularisation introduced by the replayed samples. This

difference illustrates the plasticity-stability dilemma where earlier updates favour adaptation while later updates favour retention.

**Updating Surprise During Replay.** We also considered a variant where surprise values are updated each time a sample is replayed, mimicking an aging mechanism where replaying a sample decreases its surprise, making it more likely to be replaced. This *Surprise with Updates* achieved good results (74.7% and 71.4% on the two benchmarks), but led to a decrease in performance compared to the vanilla before/after variants, suggesting that dynamic surprise is not necessary in this specific setting.

| | Standard CL Benchmark | | | | Large Number of Tasks | | | |
| | Order-1 | Order-2 | Order-3 | avg | Order-4 | Order-5 | Order-6 | avg |
|---|---|---|---|---|---|---|---|---|
| Label Surprise | 68.5 | 66.0 | 60.1 | 64.9 | 60.9 | 61.2 | 61.5 | 61.2 |
| Surprise with updates | 74.8 | 75.1 | 74.3 | 74.7 | 74.0 | 70.5 | 69.8 | 71.4 |
| Surprise Before Update After | **78.2** | 74.4 | 73.8 | 75.5 | **73.8** | **74.3** | 70.9 | 73.0* |
| Surprise Before Update Before | 77.0* | **78.1** | 76.4* | **77.2** | 72.8 | 72.1 | 71.6* | 72.1 |
| Surprise After | 76.0 | 76.7* | **76.7** | 76.5* | 73.3* | 73.0* | **73.1** | **73.1** |
| MTL | 80.0 | 80.0 | 80.0 | 80.0 | 76.3 | 76.3 | 76.3 | 76.3 |

Table 3: Final accuracy (%) on the Standard CL and the Large Number of Tasks Benchmarks for different replay variants using T5 as a base model. **Bold** indicates the best results and * is for the second best.

**Random Buffer Update After.** For a fair comparison, we evaluated our task-boundary aware Surprise Buffer against a baseline buffer that randomly selects an equal of samples per datasets, and appends them at the end of each task. We ran experiments across a wide range of buffer sizes and replay ratios which are summarised in Table 5 and 10. The results support our hypothesis that not using task identity during training harms the reservoir buffer's performance, likely due to class imbalance. Even with these additional controls, our surprise-based update rule consistently outperformed the baselines across all replay ratios. Moreover, the gains from EMA and dual LoRA heads were robustly observed in every scenario we tested.

**Buffer Size.** We also study how replay performance is impacted by the buffer size. The results summarised in Table 10 show that our Surprise Replay generally outperforms its random or reservoir alternative, with the gap increasing with the size of the buffers. The best results are obtained with the Slow Surprise After (Slow-SA) at 1500 samples (75.99%), and performance tends to improve as the buffer grows for all replay variants. While the smallest surprise buffer achieved state-of-the-art results, it does not always beat the random baseline. On the other hand, moderate sizes, 300 and 500, already capture most of the gains.

| **Buffer Size** | Random-O | Leitner-A | Random-A | Surprise-B | Surprise-A | Slow-RA | Slow-SA |
|---|---|---|---|---|---|---|---|
| 150 samples | 69.82 | 71.50 | 71.54 | 71.37 | 72.33* | **72.50** | 72.13 |
| 300 samples | 70.60 | 71.77 | 72.26 | 73.23 | 73.00 | 73.76* | **74.56** |
| 500 samples | 69.10 | 71.00 | 72.41 | 72.13 | 73.10 | 73.89* | **75.01** |
| 1500 samples | 70.70 | 71.58 | 73.04 | 73.66 | 74.58* | 73.96 | **75.99** |

Table 4: Final accuracy (%) for replay variants across buffer sizes on the LNT benchmark. Means over 3 runs × 3 task orders, replay ratio = 1:4. O, B and A indicate that the buffers are respectively updated Online, Before or After. We either add the most surprising (S) or random (R) samples.

**Replay Ratio** Finally, we fix the buffer size to 500 and vary the replay ratio, that is the number of replayed sequences per newly seen samples, from 1:2 to 1:16 (one replayed samples for every 2 or 16 new samples). As shown in Table 5, the accuracy for all methods decreases as less past samples are replayed. Here, the surprise based variants consistently outperform the random baselines, and we observe the same trend with the slow approaches. For example, at 1:2 the gains are +0.87 for the Surprise-After (Surprise-A) and +1.33 for the Slow-Surprise After (Slow-SA) compared to their random equivalent. This gap increases as the ratio is reduced, with the Surprise Before performing

best at 1:16 among the methods without EMA, and the Slow-SA outperforming the Slow Random (Slow-R) by +2.48% points. This suggests that the surprise update rule is more robust under smaller replay budgets.

| Replay Ratio | Random-O | Leitner-A | Random-A | Surprise-B | Surprise-A | Slow-RA | Slow-SA |
|---|---|---|---|---|---|---|---|
| 1:2 | 70.96 | 73.40 | 73.35 | 73.63 | 74.22 | 74.79* | **76.12** |
| 1:4 | 70.60 | 71.00 | 72.26 | 73.23 | 73.00 | 73.76* | **75.01** |
| 1:8 | 70.33 | 68.57 | 69.10 | 70.11 | 70.70 | 71.00* | **72.69** |
| 1:16 | 66.25 | 65.86 | 66.62 | 68.38* | 68.36 | 68.34 | **69.42** |

Table 5: Final accuracy (%) for replay variants across replay ratios on the LNT benchmark. Means over 3 runs × 3 task orders, buffer size = 500. O, B and A indicate that the buffers are respectively updated Online, Before or After. We either add the most surprising (S) or random (R) samples.

## 5 DISCUSSION

While our method achieves strong performance, particularly on large numbers of tasks, several limitations remain. Most significantly, our approach requires known task boundaries during training, constraining applicability to well-controlled environments, though this assumption is shared by most existing methods. Additionally, computing surprise requires an extra forward pass across datasets. Adapting our approach to fully online settings, similar to GSS and Reservoir, would address both limitations and represent a promising future direction. Further evaluation across foundation model families, e.g. LLMs like Llama (Grattafiori et al., 2024), Qwen (Yang et al., 2025a), or new modalities, Vision, Vision-Language Models, as well as settings like continual pre-training, would strengthen our findings; preliminary CPT experiments (Table 8) already show that Slow Surprise achieves the best average perplexity across domains on a small set of datasets. Finally, our dual-learner architecture shows promise and merits deeper investigation of alternative designs and training objectives.

**Neuroscience and Consolidation.** The selection–integration view mirrors core ideas in memory neuroscience. Surprise-driven selection aligns with evidence that event boundaries and prediction errors structure episodic encoding and hippocampal responses (Baldassano et al., 2017; Fountas et al., 2022; Mariola et al., 2022). In language, recent work shows that model- or behaviour-derived surprise segments narratives in ways that track human reports and neural data (Michelmann et al., 2025; Fountas et al., 2025; Benfeghoul et al., 2025). Replay is likewise thought to prioritise behaviourally valuable/surprising content, consistent with normative accounts of prioritised access and empirical biases in hippocampal replay (Mattar & Daw, 2018; Ambrose et al., 2016). Finally, EMA-style slow updates map onto complementary learning systems and multi-timescale synaptic consolidation, where fast traces are gradually integrated into stable representations (McClelland et al., 1995; Benna & Fusi, 2016). This mapping suggests concrete predictions: prioritising high-surprise sequences should preferentially protect boundary-adjacent knowledge under tight replay budgets, while removing EMA should selectively increase cross-task interference.

## 6 CONCLUSION

In this work, we revisited replay, a classical approach to catastrophic forgetting, and showed that its performance has been largely underestimated in the LLM CL literature. We then introduced *SuRe*, a surprise-based buffer update that selectively retains the most surprising samples, achieving state-of-the-art results in the *Large Number of Tasks* benchmark and delivering the best overall average performance across both *Standard CL* and *LNT* settings, with strong robustness under reduced buffer sizes and replay ratios. Our selection–integration framework explains these gains as complementary: coupling SuRe with a dual fast–slow LoRA architecture and exponential moving average (EMA) yields further improvements, including gains of up to +5 *percentage points* on LNT over prior work. These findings establish replay as a competitive and scalable baseline for continual LLM fine-tuning and highlight that jointly addressing selection and integration errors is key to mitigating catastrophic forgetting in a large number of task setting.

## 7    REPRODUCIBILITY STATEMENT

In an effort to make our work reproducible we include experimental details across section 4, an implementation details section in Appendix G as well as a proof section H to derive our claims. All the datasets we use are publicly available, either on HuggingFace or on GitHub and we are working on releasing our own public version of our codebase including all mentioned methods and ablations.

## 8    USE OF LARGE LANGUAGE MODELS

We use LLMs only for minor wording and syntactic improvement within the main text and appendices.

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

# A ADDITIONAL FIGURES

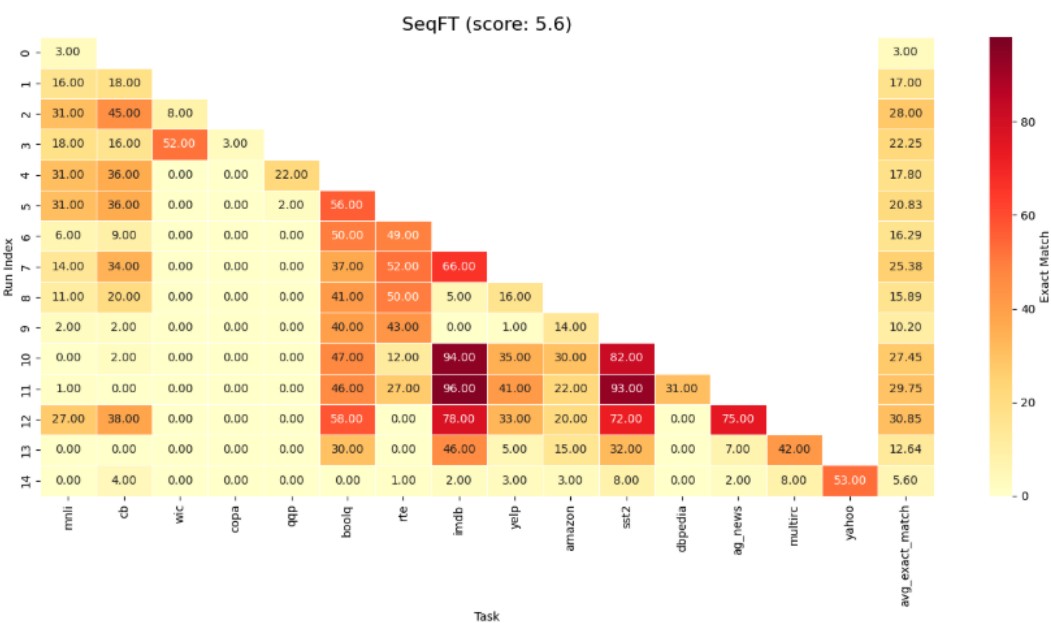

Figure 2: Naive sequential fine-tuning (SeqFT) with T5-Large on the Large Number of Tasks (LNT) benchmark.

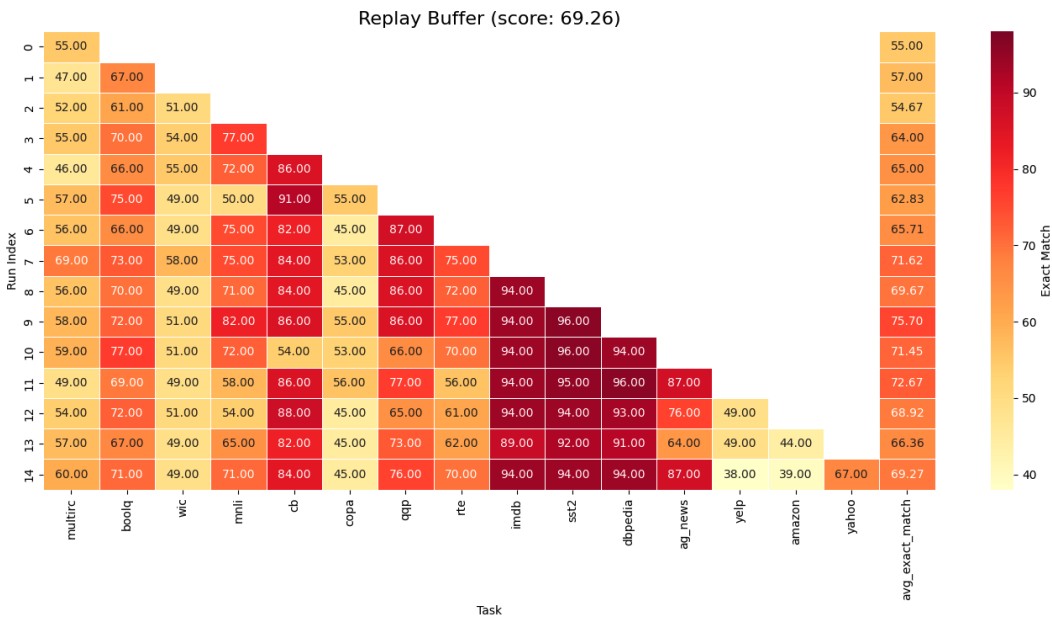

Figure 3: Reservoir Buffer replay with T5-Large on sequential tasks. Heatmap shows test task (x-axis) evaluated after each training task (y-axis).

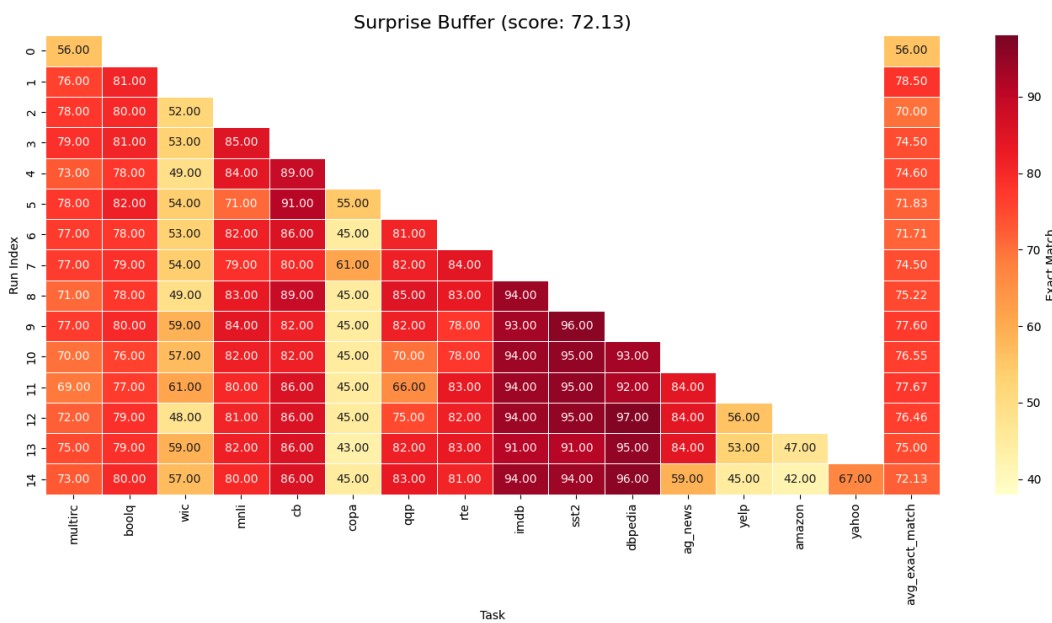

Figure 4: Surprise Buffer replay with T5-Large on sequential tasks. Same visualisation as Figure 3.

## B EXTENDED EXPERIMENTS

### B.1 MAIN RESULTS

| | **Standard CL Benchmark** | | | | **Large Number of Tasks** | | | | **All** |
|---|---|---|---|---|---|---|---|---|---|
| | Order-1 | Order-2 | Order-3 | avg | Order-4 | Order-5 | Order-6 | avg | avg |
| N-LoRA⋄ | 79.2* | 78.4 | 78.8* | 78.8* | 73.6 | 70.3 | 73.2 | 72.4 | 75.6 |
| OLieRA⋄ | **79.9** | **79.5** | **79.5** | **79.6** | 73.8 | 70.4 | 73.5 | 72.6 | 76.1 |
| *Batch 64* | | | | | | | | | |
| AimMerging | 71.2 | 72.4 | 70.9 | 71.5 | 74.1 | 73.5 | 73.7 | 73.7 | 72.6 |
| Surprise Replay | 78.4 | 77.0 | 75.8 | 77.1 | 77.6* | 76.2 | 78.0* | 77.3* | 77.2 |
| Slow Surprise Replay | 78.8 | 77.9 | 77.6 | 78.1 | **78.0** | **77.0** | **78.8** | **77.9** | **78.0** |
| *Batch 8* | | | | | | | | | |
| AimMerging | 69.5 | 70.6 | 69.2 | 69.8 | 75.9 | 74.5 | 75.9 | 75.4 | 72.6 |
| Surprise Replay | 77.9 | 77.9 | 77.1 | 77.7 | 75.8 | 74.3 | 75.4 | 75.2 | 76.5 |
| Slow Surprise Replay | 78.5 | 78.5* | 78.0 | 78.3 | 77.0 | 76.8* | 77.3 | 77.0 | 77.7* |

Table 6: Final accuracy (%) on the Standard CL and the Large Number of Tasks Benchmarks for different baselines on T5. Here the hyperparameters used are the ones reported by Cao & Wu (2025). ⋄ indicate the results were taken from Cao & Wu (2025).

### B.2 LLAMA 3.1 8B

Following Liao et al. (2025), we replicate our main setup with Llama 3.1 8B. Due to compute limits, we report only orders 1 and 4 (averaged over three runs); see Table 7. Results are preliminary but indicate that the Slow Surprise again performs strongly.

### B.3 CONTINUAL PRE-TRAINING (CPT) SETTING

We also explore Continual Pre-Training, where the model is updated via next-token prediction (no explicit labels). We select five domains from M2D2 (Reid et al., 2022), following Çağatay Yıldız et al. (2025), and report perplexity before/after various methods in Table 8. Lower is better. Our Dual

| Method | Standard CL Benchmark | LNT |
|---|---|---|
| Before FT | 1.00 | 1.00 |
| SeqFT | 23.00 | 22.86 |
| Replay | 65.75 | 50.53 |
| Slow Replay | 73.00* | 48.06 |
| Surprise Replay | 72.25 | **68.53** |
| Slow Surprise | **76.00** | 67.00* |
| Individual FT | 75.25 | 72.56 |
| MTL | 76.00 | 69.06 |

Table 7: Final accuracy (%) for Llama 3.1 8B across orders 1 and 4 (3 runs each).

Learner with Surprise Replay achieves the best average (11.63), outperforming MTL (12.01). This is a theoretical probe, downstream accuracy is not evaluated here.

| Method | Biology | Chemistry | Physical Science | Maths | Philosophy | Avg. |
|---|---|---|---|---|---|---|
| Without FT | 25.612 | 28.735 | 25.177 | 23.052 | 22.523 | 25.020 |
| SeqFT | 32.96 | 33.10 | 24.84 | 15.80 | **12.00** | 23.12 |
| Surprise Replay | 13.79 | 20.30 | 17.53 | 13.62 | 13.36 | 15.72 |
| Random Replay | 12.69 | 21.42 | 15.82 | 13.89 | 14.47 | 15.66 |
| Slow Surprise | **10.98*** | **15.47*** | **10.74** | **8.76** | 12.21* | **11.63** |
| MTL | 9.07 | 13.92 | 11.82* | 11.60* | 13.63 | 12.01* |

Table 8: Perplexity across M2D2 domains for CPT. Lower is better.

## C  MEASURING FORGETTING

We report final performance (FP), average performance (AP), and forgetting (lower is better; negative implies improvement on earlier tasks) in Table 9. The Slow Replay exhibits strong stability with negative forgetting on both benchmarks. Updating the buffer after training generally increases stability, while updating before increases plasticity.

| | Standard CL Benchmark | | | Large Number of Tasks | | |
|---|---|---|---|---|---|---|
| | AP ↑ | FP ↑ | Forget ↓ | FP ↑ | AP ↑ | Forget ↓ |
| Replay | 80.83 | 76.92 | 3.91 | 72.93 | 69.1 | 3.83 |
| Surprise Before | 78.20 | 77.20 | 1.00 | 74.80 | 72.10 | 2.70 |
| Slow Surprise | 75.80 | 78.10 | -2.30 | 70.30 | 75.10 | -4.80 |
| Surprise Before, Update After | 75.00 | 75.20 | -0.20 | 73.40 | 73.00 | 0.40 |
| Slow SB-UA | 77.00 | 77.10 | -0.10 | 70.10 | 75.00 | -4.90 |

Table 9: FP/AP/Forgetting on Standard CL and LNT benchmarks.

## D  ABLATION ON $\beta$

The parameter $\beta$ controls the integration rate of the fast weights into the slow weights (i.e. the consolidation rate of the EMA). It is thus a crucial parameter when it comes to the dual learner architecture. Our ablation suggest that too low of an integration leads to very poor performance (0.999) while lower the value (higher integration of the slow weights) leads to a less significant decrease in performance.

| $\beta$ **values** | Slow Random | Slow SB-UB | Slow SB-UA |
|---|---|---|---|
| 0.985 | 72.10 | **75.62** | 75.36* |
| 0.99 | 72.44 | **75.47** | 75.45* |
| 0.995 | 73.03 | **75.80** | 75.68* |
| 0.999 | 56.86 | **58.06** | 57.74* |

Table 10: Final accuracy (%) for slow replay variants across $\beta$ values on the LNT benchmark. Means over 3 runs × 3 task orders, replay ratio = 1:2. O, B and A indicate that the buffers are updated Online, Before and After respectively. We either add the most surprising or random samples.

## E SUM OF SURPRISE VS AVERAGE SURPRISE

Our main method relies on the average surprise per sequence to identify the most surprising labels. Here, we compare this approach with one that uses the full sequence's surprise. Our experiments indicate that the average surprise is a better indicator of importance for the replay selection.

| | **Large Number of Tasks** | | | |
|---|---|---|---|---|
| | Order-4 | Order-5 | Order-6 | avg |
| Sum Surprise | 72.82 | 72.66 | 73.03 | 72.84 |
| Average Surprise | 75.14 | 74.5 | 73.03 | 74.22 |
| Slow Sum Surprise | 75.40* | 75.78* | 75.80* | 75.66* |
| Slow Avg Surprise | **76.16** | **76.25** | **75.96** | **76.12** |

Table 11: Final accuracy (%) on the Large Number of Tasks Benchmarks for different surprise variants.

## F SURPRISE VARIANTS

We study alternative surprise computations and buffer update schedules and their trade-offs in compute, stability, and performance.

### F.1 LABEL-LEVEL SURPRISE

Instead of sequence-level surprise, compute surprise on the task label only:

$$\text{score}_i = -\log p_{\theta^{\text{pre}}}(y_i \mid x_i), \qquad R = \text{TopK}\left(\{(i, \text{score}_i)\}_{i\in D}\right), \tag{7}$$

where $x_i$ is the input, $y_i$ the label, $\theta^{\text{pre}}$ the pre-training parameters, and $R$ the retained set.

### F.2 TIMING OF SURPRISE AND BUFFER UPDATES

We vary both when surprise is computed and when the buffer is updated—before vs. after training on a dataset—yielding three variants: **SB-UB** (Before/Before), **SB-UA** (Before/After), and **SA-UA** (After/After). For a sequence $z_i$:

$$\text{score}_i = \begin{cases} S_{\theta^{\text{pre}}}(z_i), & \text{if computed before training,} \\ S_{\theta^{\text{post}}}(z_i), & \text{if computed after training,} \end{cases} \tag{8}$$

with $S_\theta(\cdot)$ the sequence-level surprise under parameters $\theta$.

### F.3 SURPRISE UPDATES DURING REPLAY

We also recompute surprise at each replay step to mimic aging:

$$\text{score}_i^{(t+1)} = S_{\theta^{(t)}}(z_i), \qquad \text{with score}_i^{(t+1)} \leq \text{score}_i^{(t)} \text{ as training progresses.} \tag{9}$$

## G  IMPLEMENTATION DETAILS

Hyperparameters follow Wang et al. (2023) unless noted: learning rate $1e{-}3$ (T5-Large) and $1e{-}4$ (Llama 3.1 8B); batch size 64; replay frequency $1/2$ (every other gradient step); one epoch; dropout 0.1; LoRA rank 8 and $\alpha = 32$; LoRA adapters on $Q$ and $V$ projections in all attention layers.

## H  PROOFS AND TECHNICAL DETAILS

We restate our local assumptions in the LoRA parameter subspace (base weights frozen):

(A1) **Local smoothness/PL:** Each task risk $R_k(\theta) = \mathbb{E}_{z \sim P_k} \ell(\theta; z)$ is $L$-smooth and satisfies a local $\mu$-PL inequality on the trajectory neighborhood $\mathcal{N}$: for some $\mu > 0$, $\frac{1}{2}\|\nabla R_k(\theta)\|^2 \geq \mu\big(R_k(\theta) - R_k(\theta_k^\star)\big)$ for all $\theta \in \mathcal{N}$. Per-example gradients are bounded: $\|\nabla_\theta \ell(\theta; z)\| \leq G$.

(A2) **Stochastic optimisation:** The fast learner performs SGD on $J_t(\theta) = (1 - \alpha)R_t(\theta) + \alpha\,\tilde{R}_{1:t-1}(\theta)$ with step $\eta \leq 1/L$: $\theta_f^{(n+1)} = \theta_f^{(n)} - \eta\, g^{(n)}$, where $\mathbb{E}[g^{(n)} \mid \theta_f^{(n)}] = \nabla J_t(\theta_f^{(n)})$ and $\mathbb{E}\|g^{(n)} - \nabla J_t(\theta_f^{(n)})\|^2 \leq \sigma^2$. The slow learner is EMA: $\theta_s^{(n+1)} = \beta \theta_s^{(n)} + (1-\beta)\theta_f^{(n+1)}$, $\beta \in (0, 1)$.

(A3) **Task drift:** $\|\theta_{k+1}^\star - \theta_k^\star\| \leq \delta$.

We use standard facts about SGD stability and Polyak–Ruppert averaging (Polyak & Juditsky, 1992; Konda & Tsitsiklis, 2004; Borkar, 2008; Hardt et al., 2016) and integral probability metrics (IPMs; MMD is a special case) (Gretton et al., 2012).

*Remark.* In the main text we allow a generic consolidation operator $\mathcal{A}_\psi$. In this appendix we instantiate $\mathcal{A}_\psi$ as EMA with parameter $\psi \equiv \beta$, hence bounds are stated with $B(\beta)$; this corresponds to $B(\psi)$ in Theorem 1.

### H.1  PROOF OF LEMMA 1 (SELECTION MISMATCH VIA IPM)

Recall $P_{1:t-1} = \frac{1}{t-1}\sum_{k<t} P_k$, $\tilde{R}_{1:t-1}(\theta) = \mathbb{E}_{z \sim q}\ell(\theta; z)$ and $R_{1:t-1}(\theta) = \mathbb{E}_{z \sim P_{1:t-1}}\ell(\theta; z)$. Let $\mathcal{F}_{\text{loc}} = \{\ell(\theta; \cdot) : \theta \in \mathcal{N}\}$ be the set of per-example losses reachable along the local trajectory. An integral probability metric $D_{\mathcal{F}_{\text{loc}}}$ is defined by

$$D_{\mathcal{F}_{\text{loc}}}(P, Q) = \sup_{f \in \mathcal{F}_{\text{loc}}} \left| \mathbb{E}_P f - \mathbb{E}_Q f \right|.$$

For any fixed $\theta \in \mathcal{N}$, take $f_\theta(\cdot) = \ell(\theta; \cdot) \in \mathcal{F}_{\text{loc}}$. Then

$$\left| \tilde{R}_{1:t-1}(\theta) - R_{1:t-1}(\theta) \right| = \left| \mathbb{E}_{z \sim q}\ell(\theta; z) - \mathbb{E}_{z \sim P_{1:t-1}}\ell(\theta; z) \right| \leq D_{\mathcal{F}_{\text{loc}}}\big(P_{1:t-1}, q\big).$$

This is exactly Eq. equation 1. $\blacksquare$

**Remark (MMD instance).** If $D_{\mathcal{F}_{\text{loc}}}$ is the RKHS IPM (MMD) for a kernel $k$, then for any function class embedded in that RKHS one gets $\left| \tilde{R} - R \right| \leq \|\ell(\theta; \cdot)\|_{\mathcal{H}} \cdot \text{MMD}_k(P_{1:t-1}, q)$. We keep the abstract IPM to avoid extra regularity assumptions on $\ell(\theta; \cdot)$.

### H.2  PROOF OF LEMMA 2 (EMA REDUCES INTEGRATION VARIANCE)

We analyse the EMA of fast SGD iterates on $J_t$ in a local basin containing a unique PL stationary point $\theta^\star$. Define the fast error $e^{(n)} = \theta_f^{(n)} - \theta^\star$ and the slow (EMA) average

$$\bar{\theta}_N := (1 - \beta)\sum_{n=1}^{N} \beta^{N-n} \theta_f^{(n)}, \qquad \bar{e}_N := \bar{\theta}_N - \theta^\star = (1 - \beta)\sum_{n=1}^{N} \beta^{N-n} e^{(n)}.$$

**Step 1: linearised SA recursion.** By $L$-smoothness and PL near $\theta^\star$, the fast recursion linearises to

$$e^{(n+1)} = (I - \eta H)e^{(n)} + \eta\xi^{(n)},$$

where $H := \int_0^1 \nabla^2 J_t\big(\theta^\star + s(\theta_f^{(n)} - \theta^\star)\big)\,ds$ satisfies $H \succeq \mu I$ and $\|I - \eta H\| \leq (1 - \eta\mu)$ for $\eta \leq 1/L$. The noise $\xi^{(n)} := g^{(n)} - \mathbb{E}[g^{(n)} \mid \theta_f^{(n)}]$ is a martingale-difference with $\mathbb{E}\|\xi^{(n)}\|^2 \leq \sigma^2$.

**Step 2: EMA as low-pass / Polyak–Ruppert averaging.** Classical two-time-scale/averaging arguments (e.g., Polyak & Juditsky, 1992; Konda & Tsitsiklis, 2004; Borkar, 2008) imply a decomposition of the EMA mean-square error into a *bias* term (how far the average lags the trackable optimum) and a *variance* term (how noise is filtered):

$$\mathbb{E}\|\bar{e}_N\|^2 \ \leq \ C_1\,(1-\beta)^2\,\|e^{(0)}\|^2 \ + \ C_2\,\frac{1}{(1-\beta)}\,\frac{\sigma^2}{\mu N},$$

for universal constants $C_1, C_2$ depending on $L$ and the spectral gap of $H$; see, e.g., Theorem 1 in Polyak & Juditsky (1992) and Theorem 2.2 in Konda & Tsitsiklis (2004) adapted to geometrically weighted averages (EMA).

Intuition: EMA is a geometrically weighted average with effective window length $\approx 1/(1-\beta)$; averaging reduces variance by the window length (hence the $1/(1-\beta)$ factor) while incurring a steady-state bias proportional to the leakage $(1-\beta)$.

**Step 3: risk bound under smoothness/PL.** Using $L$-smoothness of $R_k$ and Jensen,

$$\mathbb{E}\big[R_k(\bar{\theta}_N) - R_k(\theta_k^\star)\big] \ \leq \ \tfrac{L}{2}\,\mathbb{E}\|\bar{e}_N\|^2 \ + \ C_d\,\delta,$$

where $C_d\,\delta$ accounts for bounded drift between $\theta^\star$ (minimiser of $J_t$) and $\theta_k^\star$ (minimiser of $R_k$) across successive tasks (Assumption (A3)). Substituting the EMA MSE bound yields

$$\mathbb{E}\big[R_k(\bar{\theta}_N) - R_k(\theta_k^\star)\big] \ \leq \ C_b\,(1-\beta) \ + \ C_v\,\frac{1}{(1-\beta)}\,\frac{\sigma^2}{\mu N} \ + \ C_d\,\delta,$$

which is Eq. equation 2. ∎

### H.3 Proof of Theorem 1 (Additive bound; complementary knobs)

Fix any past task $i < t$ and consider one training phase over task $t$. We compare the *slow* model before and after the phase. Let $\bar{\theta}^{\mathrm{pre}}$ and $\bar{\theta}^{\mathrm{post}}$ denote the slow (EMA) parameters at the start and end of the phase, and let $\theta_f^{\mathrm{pre}}, \theta_f^{\mathrm{post}}$ be the corresponding fast parameters at those times.

Decompose the change in $R_i$ over the phase as

$$R_i(\bar{\theta}^{\mathrm{post}}) - R_i(\bar{\theta}^{\mathrm{pre}}) = \underbrace{R_i(\bar{\theta}^{\mathrm{post}}) - R_i(\theta_f^{\mathrm{post}})}_{\text{(A) fast}\to\text{slow (variance)}}$$
$$+ \underbrace{R_i(\theta_f^{\mathrm{post}}) - R_i(\theta_f^{\mathrm{pre}})}_{\text{(B) fast drift over the phase}}$$
$$+ \underbrace{R_i(\theta_f^{\mathrm{pre}}) - R_i(\bar{\theta}^{\mathrm{pre}})}_{\text{(C) slow}\to\text{fast (variance)}}.$$

**Term (A)+(C): variance controlled by EMA.** By Lemma 2, both differences between fast and slow parameters can be bounded in expectation by the EMA bias/variance expression:

$$\mathbb{E}\big[(A) + (C)\big] \ \leq \ C_b(1-\beta) + C_v\,\frac{1}{(1-\beta)}\,\frac{\sigma^2}{\mu N} \ + \ C_d\,\delta.$$

**Term (B): slow drift driven by mixed gradients and selection bias.** The fast drift over the phase is driven by SGD on $J_t$; replacing the replay risk $\tilde{R}_{1:t-1}$ by the true past risk $R_{1:t-1}$ introduces a bias *per step* controlled by the IPM gap (Lemma 1):

$$\big|\tilde{R}_{1:t-1}(\theta) - R_{1:t-1}(\theta)\big| \ \leq \ D_{\mathcal{F}_{\mathrm{loc}}}\big(P_{1:t-1}, q\big) \quad \text{for all } \theta \in \mathcal{N}.$$

Standard stability arguments for SGD on $L$-smooth losses (e.g., Hardt et al., 2016) imply that replacing the objective by a uniformly $\varepsilon$-perturbed one perturbs the risk along the trajectory by at most a constant multiple of $\varepsilon$ (over a finite number of steps in the local region). Thus

$$\mathbb{E}\big[(B)\big] \ \leq \ A \cdot D_{\mathcal{F}_{\mathrm{loc}}}\big(P_{1:t-1}, q\big) \ + \ C_d\,\delta,$$

for some $A$ depending on $L$ and the phase length.

**Summing over phases.** Summing (A)+(B)+(C) across all phases/tasks up to $T$ and averaging over $i < T$ yields

$$\mathbb{E}\,\mathcal{F} \;\le\; A\,D_{\mathcal{F}_{\mathrm{loc}}}\big(P_{1:T-1}, q\big) \;+\; B(\beta)\,\frac{\sigma^2}{\mu N} \;+\; C\,\Delta_{\mathrm{drift}},$$

which is Eq. equation 3 with $B(\psi) \equiv B(\beta)$ for EMA. Since $m < \infty$ (finite memory) implies $\inf_q D_{\mathcal{F}_{\mathrm{loc}}}(P, q) > 0$ and $N < \infty$ with $\beta < 1$ implies $\frac{1}{(1-\beta)}\frac{\sigma^2}{\mu N} > 0$, neither addend can be driven to zero by tuning the other; therefore the buffer policy (selection) and EMA (integration) are complementary controls. ∎

**On surprise-based selection.** A general proof that sequence-level surprise *minimises* $D_{\mathcal{F}_{\mathrm{loc}}}(P, q)$ at fixed memory would require extra structural assumptions on $\ell(\theta; \cdot)$ and the data distribution. Instead, we motivate it via importance sampling (high-loss/high-gradient points reduce estimator variance (Zhao & Zhang, 2015; Katharopoulos & Fleuret, 2018)) and validate empirically in our main experiments.

