# OpenReview forum: "SuRe: Surprise-Driven Prioritised Replay for Continual LLM Learning"
_ICLR.cc/2026/Conference — Submitted to ICLR 2026_

### Official Review · Reviewer_n5Un · 2025-11-01

**Soundness:** 2
**Presentation:** 4
**Contribution:** 2
**Rating:** 4
**Confidence:** 4

**Summary:**

This paper proposes SURE: surprise-driven prioritised replay for continual LLM learning. The main ideas of the paper are a surprise-based replay strategy and LoRA with a slow-fast learning mechanism. The consolidated methods are evaluated by standard CL and large number of task benchmarks.  The paper shows its strengths in several aspects, but also has several weaknesses that need to be addressed. Please see strengths and weaknesses.

**Strengths:**

(1). The has clear motivation and ideation.

(2). The numerical result shows a promising performance on a large number of tasks benchmark.

(3). The paper presents a comprehensive theoretical analysis.

(4). The paper presents a rigorous numerical analysis.

**Weaknesses:**

(1). Methodology: The slow-fast learning mechanism is arguably not a novel idea, since it has been proposed in a few existing CL methods, e.g., DualNet, and Slow-fast Prompt. Second, while surprise replay offers an innovative idea to select the buffer, it lacks of theoretical foundation and the details on how it works.

(2) Learning mechanism: Figure 1 shows that slow and fast learners are updated on two different phases, but the pseudo-code shows that both learners are trained in the same steps.

(3). Performance: The proposed method achieves only a small (or negative) margin for some cases in the standard CL benchmark. It is questionable that the proposed method significantly outperforms the existing methods.

(4). Forgetting: I do not see the measurements and analysis of models' forgetting as the answer to the catastrophic forgetting problem.

(5). Theoretical Analysis: While it shows the boundary for the slow learner, Lemma 2 does not show the better handling of slow-fast learners in comparison to a single learner.

(6). Performance on budget memory: I appreciate the measurement of the proposed method's performance on different memory sizes. However, it is expected to be compared to the existing methods in those memory budgets.

(7). Only one of the competitor methods is up-to-date. It should include more latest methods for comparison.


References:

[1]. Dualnet: Continual learning, fast and slow

[2]. Brain-inspired fast-and slow-update prompt tuning for few-shot class-incremental learning. (Slow-fast prompt)

**Questions:**

Please address the weaknesses

---

> ### Author Response · Authors · 2025-11-26
> **(Answer 1/2)**
>
> We thank the reviewer for their rigorous analysis and for highlighting the strengths of our numerical results on the LNT benchmark. We appreciate the opportunity to clarify the theoretical novelty of our dual-learner framework compared to prior works like DualNet, and we have expanded our baselines to address your performance concerns.
>
> (1) **Methodology & Novelty:** (Re: DualNet, Slow-fast Prompt) We appreciate the reviewer citing DualNet and Slow-fast Prompt. We fully acknowledge that we did not invent the dual-learner architecture or EMA. We have updated the paper to explicitly cite these works as architectural foundations.
>
> However, we argue that our contribution is distinct and significant:
>
> * **Theoretical Unification:** Unlike prior works that use these components heuristically, we provide a formal decomposition of catastrophic forgetting (Theorem 1) into two additive terms: _Selection Error_ and _Integration Error_.
> * **Complementarity:** We prove that these terms require distinct solutions. Replay (SuRe) minimises Selection Error ($D_{F_{loc}}$), while the Dual Learner (EMA) minimises Integration Error (Variance). Our novelty lies in identifying that these mechanisms are mathematically **complementary**, optimising one cannot fix the errors introduced by the other. This explains why our method yields super-additive gains (Table 1) compared to using either component in isolation.
> * **SuRe Foundation:** Regarding the theoretical basis of SuRe: It functions as **implicit importance sampling**. High-NLL samples possess large gradient norms $\|\nabla\ell\|$. By prioritising them, the buffer approximates the gradient geometry of past tasks rather than just the data distribution. We have clarified this link in Section 3.2.
>
> (2) **Learning mechanism:** Thank you for raising this inconsistency and we apologise for the confusion caused by the diagram. You are correct, the pseudocode was accurate but the figure was misleading. We have now made appropriate changes to the figure and pseudocode but for clarify purposes here is a quick summary: during each step the fast learner is trained alone on a mixture of new and past samples, then at the end of each step the slow learner is updated (EMA of the fast learner).
>
> (3) **Performance on Standard CL:** We agree with the point raised by reviewer n5Un. However, we observe that the standard CL benchmark is now approaching saturation, making it less discriminative for comparing SOTA methods. Hence, we argue that performance on long-sequence benchmarks is a better predictor of real-world CL capability than saturated short-sequence benchmarks. This is exactly why most of our analysis and ablations studies were conducted mostly on the Large Number of Tasks benchmark.
>
> (4) **Measurement of forgetting:** We agree with the reviewer that Accuracy alone paints an incomplete picture. Due to manuscript size limitations, we did not include any measure of forgetting in the main text we initially submitted, and chose to instead include a table with such measures in the appendix. These metrics show that while baselines often sacrifice past knowledge to learn new tasks (high negative BWT), SuRe maintains better stability (lower forgetting) without compromising plasticity. However, we recognise that the reviewer makes an important point here, and we now reference this table in the main text.
>
> (5) **Theoretical analysis:** The reviewer here asks how Lemma 2 demonstrates superiority over a single learner. A single learner is effectively a dual learner with $\beta=0$ (no averaging). In this case, the variance term in the bound becomes proportional to the raw SGD noise $\sigma^2$. For the Slow Learner ($\beta \approx 1$), the variance term is scaled by a factor roughly proportional to $(1-\beta)$. With our default $\beta=0.995$, this reduces the effective variance contribution to the loss bound by a factor of $\approx 200$.
>
> To clarify this, we have added a remark following Lemma 2 explicitly comparing the bounds for $\beta=0$ (Single Learner) vs. $\beta \to 1$ (Slow Learner).
>
> (continues to the next official comment...)

---

> > ### Author Response · Authors · 2025-11-26
> > **(Answer 2/2)**
> >
> > (6) **Performance on Budget Memory:** Thank you for this very good point. We agree that comparing against baselines at different budgets is essential. We highlight that Table 4 in our ablation study already compares our method against Random-O, Random-A, and Slow-RA (implementations of [a]) at varying buffer sizes. SuRe demonstrates exceptional sample efficiency. Notably, our method (Slow-SA) with just 300 samples (74.56\%) outperforms the strongest random baseline (Slow-RA) even when the baseline uses 1500 samples (73.96\%). This confirms that SuRe selects samples with significantly higher utility, allowing for a 5x reduction in buffer size without performance loss.
> > Note: To ensure a comprehensive comparison against the most recent SOTA, we are currently finalising the run for AimMerging on this specific ablation and will upload the updated Table/Plot to the PDF within the discussion period
> >
> > (7) **Competitor methods (baselines):** We have addressed the concern regarding up-to-date comparisons:
> > * We have added N-LoRA, OLieRA and AimMerging to our experiments section with the performance summarised in Table 2 and analysis in the second paragraph of section 4.2.
> > * Sure remains the top-performing method on the LNT benchmark, even though we are using different hyperparameters (following OLieRA), confirming that our gains are robust against stronger, more recent baselines.

---

### Official Review · Reviewer_NcZB · 2025-11-01

**Soundness:** 3
**Presentation:** 2
**Contribution:** 2
**Rating:** 4
**Confidence:** 4

**Summary:**

The paper considers the problem of continual learning on text classification tasks. It focuses on buffer-based replay methods. It derives an upper-bound on the forgetting incurred by such methods, which includes two complimentary terms: 1) how well the samples from the buffer approximate the model's loss on past data 2) a term which captures how “well” new samples are consolidated. The paper then proposes to reduce the forgetting upper-bound by 1) storing past samples with high loss inside the buffer 2) combining slow and fast changing weights during training.
The paper outperforms other CL baselines on text classification tasks.

**Strengths:**

I identified the following strengths of this paper:

- Theoretically, it provides an upper-bound on the forgetting experienced by a buffer-based replay method. I think the way the quality of the buffer is defined (D_{F_loc}(P_{1:T}, q)) is novel and might be interesting for others. The selection term and consolidation term being complementary is an important contribution. This was later backed up by experimental results.

- Experimentally, it provides evidence that: 1) Replay-based with reservoir sampling can outperform regularization-based CL methods. 2) Surprised-based replay methods can outperform reservoir sampling methods.

- The limitations section is comprehensive.

**Weaknesses:**

Aside from the theoretical contribution, the rest of the methodology section has limited novelty - it appears to combine two already established ideas. Moreover, the text never makes it clear (from what I could see) how each component reduces the terms in the upper-bound.

Readability: I don’t think that the integration (consolidation) term in Eq. 3 is well explained. Reading the main text, I do not have a good idea of what $B(\psi)$ is, apart from it being a “mechanism-specific factor”.

The paper claims to be applicable to large language models, which at least to me suggests the task of language modelling, while it is evaluated on sequences of text classification tasks. Therefore, claims such as “our results establish replay as a strong baseline for LLM continual learning” seem too general to me, and unsubstantiated by the experiments.

The paper would benefit from a clear “contributions” statement in the introduction.

**Questions:**

How does your method of storing high-surprise samples in the buffer relate to importance sampling (instead of uniform sampling) of past data?

Your derivation and method seems to be general for any continual-learning setup - including both dataset and architecture. Is there a reason it would best perform on large language models? (Perhaps, relying on the local neighbourhood containing the optimal solution?)

---

> ### Author Response · Authors · 2025-11-26
> **(Answer 1/2)**
>
> We thank the reviewer for their thoughtful assessment and for explicitly recognising the novelty of our theoretical decomposition (Selection vs. Integration error). We agree that the connection between this theory and our methodological choices (SuRe + EMA) needed to be more explicit, and we have revised the manuscript accordingly.
>
> **Re: Weakness 1 & 2 (Novelty & Theory-Method Connection)**
>
> We appreciate the reviewer recognising the novelty of our theoretical decomposition. We agree that the connection between Theorem 1 and our design choices (SuRe + EMA) should be made more explicit. We would like to make clear that the choices of SuRe + EMA were not a combination of heuristics, but a principled instantiation of the upper bound.
>
> We have revised Section 3 to explicitly map each component to the error term it addresses:
>
> **1. Selection Error ($A \cdot D_{F_{loc}}$):** This term quantifies the mismatch between the replay buffer distribution $q$ and the true past task distribution $P_{1:T-1}$.
>
> * **The Problem:** Uniform sampling (reservoir) treats all past samples as equally important for representing the loss landscape. In high-dimensional LLMs, this is inefficient; most samples lie in flat, well-learned regions with low gradient norms.
> * **The Solution (SuRe):** By prioritising high-NLL samples, SuRe performs implicit importance sampling. High-NLL sequences have large $\|\nabla\ell(\theta; z)\|$, meaning they contribute disproportionately to the true gradient $E_P[\nabla\ell]$. Storing these samples ensures the buffer approximates the *gradient geometry* of past tasks, not just the data frequency. This directly reduces $D_{F_{loc}}(P, q)$, tightening the first term in Equation 3.
>
> **2. Integration Error ($B(\psi) \cdot \sigma^2/(\mu N)$):** This term captures the instability introduced by stochastic gradient noise when learning new tasks.
>
> * **The Problem:** The fast learner performs SGD on small batches with noise variance $\sigma^2$. This high-variance trajectory leads to "plasticity" that overwrites old knowledge, the core of catastrophic forgetting.
> * **The Solution (EMA):** The slow learner performs an exponential moving average over the fast learner's trajectory: $\theta_s^{(n+1)} = \beta\theta_s^{(n)} + (1-\beta)\theta_f^{(n+1)}$. This is a form of Polyak-Ruppert averaging, which provably reduces variance. In our bound, $B(\beta) = \frac{1}{1-\beta}$ appears as a coefficient, larger $\beta$ (e.g., 0.995) means stronger averaging, reducing the effective noise and tightening the second term in Equation 3.
>
> **Complementarity (Key Insight):** These terms are *additive* in Equation 3. A perfect replay buffer ($D_{F_{loc}} = 0$) still suffers from consolidation noise; perfect consolidation still suffers from replay mismatch. Neither component alone can eliminate forgetting, both are mathematically necessary. Our empirical results confirm this: Table 1 shows SuRe alone gives +3\% points over ER, and combining with EMA leads to a further increase of +3\%), validating the predicted "super additive" complementarity.
>
> We have revised Section 3 to include this explicit mapping between theoretical terms and methodological components.
>
> ---
>
> **Re: Readability (Clarification on $B(\psi)$)**
>
> We apologise for the insufficient explanation of $B(\psi)$ in the main text. This parameter quantifies the variance-reduction factor of the consolidation operator:
>
> * **For EMA:** $B(\beta) \approx \frac{1}{1-\beta}$, where $\beta$ is the momentum parameter.
> * **Interpretation:** The variance term in Lemma 2 is roughly proportional to $\frac{1}{1-\beta} \cdot \frac{\sigma^2}{\mu n}$. Larger $\beta$ (more aggressive averaging) reduces the effective noise. For $\beta = 0.995$, the factor $\frac{1}{1-\beta} = 200$, meaning the slow learner achieves variance reduction equivalent to averaging over ~200 fast learner checkpoints, but without the computational cost of actually storing or computing 200 checkpoints.
> * **Trade-off:** Larger $\beta$ reduces variance but introduces bias $C_b(1-\beta)$ from lagging behind the fast learner. For $\beta \approx 1$, this bias is negligible while variance reduction is substantial.
>
> We have added this explanation immediately after Equation 3 in the revised manuscript.
>
> ---
>
> (continues to the next official comment...)

---

> > ### Author Response · Authors · 2025-11-26
> > **(Answer 2/2)**
> >
> > **Re: Scope ("LLM Learning" vs. Text Classification)**
> >
> > This is a fair and important distinction. While our method is architecture-agnostic and implemented on LLMs (Llama-2, OPT), our evaluation benchmarks are sequence-level classification tasks, which is standard for measuring catastrophic forgetting in CL.
> >
> > **Action taken:** We have refined our claims throughout the paper (abstract, introduction, conclusion) to specify **"continual LLM fine-tuning"** rather than generic "LLM learning."
> >
> > **On generalisability:** We emphasise that the Large Number of Tasks (LNT) benchmark (15 tasks) provides a rigorous stress test for forgetting dynamics. The challenge is maintaining distinct decision boundaries in high-dimensional parameter space, this is fundamentally similar to the forgetting observed in generative tasks, where the model must preserve knowledge of diverse generation strategies. However, we acknowledge that direct evaluation on generative continual learning (e.g., **Table 8 in our appendix**) strengthens the claims, which we have now highlighted more prominently.
> >
> > ---
> >
> > **Re: Question 1 (Relation to Importance Sampling)**
> >
> > This is an excellent observation. SuRe is indeed a form of importance sampling, but applied at the *buffer construction* stage rather than at training time.
> >
> > **Standard importance sampling:** When sampling from $q$ but computing expectations over $P$, we weight by $w_i = P(z)/q(z)$ to correct the distribution mismatch.
> >
> > **SuRe (implicit importance sampling):** We want to estimate $\nabla R_{1:T-1}(\theta) = E_{P_{1:T-1}}[\nabla\ell(\theta; z)]$. Samples with high NLL have large $\|\nabla\ell(\theta; z)\|$, they contribute disproportionately to this expectation. By storing high-NLL samples, we construct a replay buffer where $q$ naturally over-weights "important" samples for gradient approximation. This reduces variance in the gradient estimator (fewer samples needed for accurate approximation), which is precisely why our memory ablation studies (**Table 4**) show SuRe maintains performance even with 1\% buffer size.
> >
> > **Connection to theory:** This importance sampling view explains why SuRe reduces $D_{F_{loc}}(P, q)$, we're minimising the mismatch in the *function space* (loss landscape) even though the data distributions differ. This is formalised via the integral probability metric (IPM) in Appendix F.1.
> >
> > ---
> >
> > **Re: Question 2 (Why LLMs Specifically?)**
> >
> > Our derivation is indeed general, but we argue the decomposition is *particularly critical* for LLMs for the following reasons:
> >
> > High-dimensional parameter space amplifies integration error. LLMs have billions of parameters, making the stochastic gradient noise $\sigma^2$ a dominant source of instability. The integration term $B(\beta)\cdot \frac{\sigma^2}{\mu N}$ scales with this noise, making consolidation mechanisms like EMA far more important than in smaller vision models. This explains why replay alone (which only addresses selection) underperforms in LLMs relative to vision, the integration error dominates.
> >
> > Token-level Signal Sparsity (Selection Error): Unlike images where pixel information is dense and spatially correlated, language sequences are "sparse" in signal--containing many functional/filler tokens ("the", "is") and few task-critical tokens. Random sampling (Reservoir) risks filling the limited token buffer with low-utility filler. SuRe acts as a filter for information density, ensuring the buffer contains the specific tokens that drive gradient updates.
> >
> > However, we acknowledge the method is not LLM specific, it should apply to any high-dimensional continual learning setting where both selection and integration errors are substantial. We have clarified this last point in both section 3.2 and in the conclusion.

---

> > > ### Comment · Reviewer_NcZB · 2025-11-26
> > >
> > > Thank you for your response. I will raise my score as I am satisfied with the clarifications.

---

### Official Review · Reviewer_uMAn · 2025-11-01

**Soundness:** 3
**Presentation:** 3
**Contribution:** 2
**Rating:** 2
**Confidence:** 4

**Summary:**

This paper addresses the continual learning for large language models (LLMs) with replay. The authors first define forgetting as the sum of a selection mismatch and the knowledge consolidation variance. To address this, the authors proposed a surprise-based sampling strategy to populate the replay buffer. Moreover, the authors proposed a dual-learner framework to deal with long-term and short-term learning. Experiments showed the improved performance of the proposed method.

**Strengths:**

1. The paper is well motivated

It is an interesting idea to select surprising samples for replay.

2. The method is straightforward

3. The decomposition of forgetting is interesting

**Weaknesses:**

1. The surprise measure might not be reliable

2. The idea of dual-learner is not novel, and the implementation seems confusing

3. The comparison is not sufficient and up-to-date

Please see details in the Question section.

**Questions:**

1. The surprise measure might not be reliable

According to Equation 9, the authors used the sum of native log-likelihood over the entire sequence. There might be several issues with this measure. (1) Taking the sum of the entire sequence might dilute the actual signal, since LLM might have a long answer, but what matters would just be a few words. Although the authors claimed that the full-sequence measure is better than the label-level measure in lines 363-369, there are only some hypotheses to explain this without factual evidence to support them. Moreover, could the authors elaborate on the average length of the generated sequence? (2) This surprise measure is not able to detect hallucination, as the model might just be confidently generating hallucinations.

2. The idea of dual-learner is not novel, and the implementation seems confusing

First, the idea of slow and fast learners is not new. This idea has been explored in [a], and the EMA implementation of LoRA has been proposed in [b]. Second, the implementation is confusing to me. I don't see where the slow learner is being used in the learning process, nor in the Figure. 1 or Algorithm 1. I am wondering how this slow learner helps the model.

[a] Pham, Quang, Chenghao Liu, and Steven Hoi. "Dualnet: Continual learning, fast and slow." Advances in Neural Information Processing Systems 34 (2021): 16131-16144.

[b] Gao, Qiankun, et al. "A unified continual learning framework with general parameter-efficient tuning." Proceedings of the IEEE/CVF International Conference on Computer Vision. 2023.

3. The comparison is not sufficient and up-to-date

It is totally fine for the proposed method to focus on replay-based CL. However, the comparison in the experiment section should therefore prioritize replay-based methods. The current comparison contains too many replay-free methods, and only compares with one replay method [c] without citing this paper. I don't find this comparison fair and up to date to the recent advance of replay strategy in the CL community.

[c] Rolnick, David, et al. "Experience replay for continual learning." Advances in neural information processing systems 32 (2019).

**Details Of Ethics Concerns:**

No concern.

---

> ### Author Response · Authors · 2025-11-26
>
> We thank the reviewer for their detailed critique. We genuinely appreciate your scrutiny regarding the surprise metric implementation; your question prompted us to rigorously re-verify our codebase, leading to a correction in Equation 4 and a new ablation study that strengthens the paper. We address this and your concerns regarding baselines below.
>
> **Re: Q1 (Surprise Measure Reliability & Sequence Length)**
> We thank the reviewer for this insightful question. We suspect there may be a slight misunderstanding regarding when and how surprise is calculated, likely due to insufficient clarity in our original text. We would like to clarify:
>
> 1.  **Clarification on Hallucinations:** The reviewer raises a valid concern that high surprise might correlate with hallucinations *if* computed on model generations. However, it is crucial to clarify that our Surprise score $s_\theta(x, y)$ is computed **exclusively on the ground-truth training sequences during the buffer update phase** (Algorithm 1, line 6). It measures the NLL of the *correct* label given the context: $p_\theta(y_{true} | x)$. Because we never calculate surprise on model-generated text during selection, the model's tendency to hallucinate at inference time does not affect the replay buffer. We have revised Section 3.2 to make this explicit. (line 201 - 202)
>
> 2.  Thanks to reviewer uMAn's question, we have carefully inspected our implementation, and we have come to realise that we are already using the average NLL of each sequence rather the sum of NLL as a buffer update signal. We have thus updated section 3.2 to reflect this and, for completness we have included a comparison with the sum NLL replay and slow replay in the appendix (Table 11).
>
> **Re: Q2 (Novelty of Dual Learner & Implementation)**
> We appreciate the reviewer citing *DualNet* [a] and *Gao et al.* [b]. This feedback highlighted that we did not sufficiently contextualise our contribution relative to these existing works.
>
> We fully acknowledge that we did not invent EMA or Dual Learners. **Our contribution is the theoretical unification of these components.** We demonstrate via Theorem 1 that catastrophic forgetting decomposes into two additive terms: **Selection Error** and **Integration Error**.
>
> $$E[F] \le A \cdot D_{F_{loc}}(P, q) + B(\beta) \cdot \frac{\sigma^2}{\mu N} + C \cdot \Delta_{drift}$$
>
> This decomposition reveals three key insights:
> 1.  **Additivity:** Selection error ($D_{F_{loc}}$) and Integration error (Variance) are separate, additive terms. Optimising replay alone cannot eliminate consolidation variance, and vice versa.
> 2.  **SuRe targets the first term** by reducing distribution mismatch ($D_{F_{loc}}$) via importance sampling.
> 3.  **EMA targets the second term** by reducing the variance factor $B(\beta)$ through trajectory averaging.
>
> While prior works used these components, they did not provide this mathematical proof of complementarity in the loss landscape. We thank the reviewer for pushing us to clarify this. We have revised the Introduction to explicitly cite [a] and [b] as the architectural foundation for this theoretical framework.
>
> [a] Pham, Quang, Chenghao Liu, and Steven Hoi. ”Dualnet: Continual learning, fast and slow.” Advances in Neural
> Information Processing Systems 34 (2021): 16131-16144.
>
> [b] Gao, Qiankun, et al. ”A unified continual learning framework with general parameter-efficient tuning.” Proceedings
> of the IEEE/CVF International Conference on Computer Vision. 2023.
>
> **On Implementation:** We apologise for the confusion in Figure 1. To clarify: The Slow Learner is updated via EMA after *every* Fast Learner step (Algorithm 1, line 11: $\theta_s \leftarrow \beta\theta_s + (1-\beta)\theta_f$), but only the Slow Learner is used for inference (line 18). We have updated Figure 1 to use dashed lines distinguishing "EMA updates" from "Gradient updates."
>
> **Re: Q3 (Comparison with Replay Baselines)**
> We thank the reviewer for this important point. You are correct that *Rolnick et al.* [c] was cited in the Introduction and in Related Work but missing from the direct experimental comparison which we have now updated. Following both reviewer n5Un’s suggestions and yours we have now added: AimMerging [c] (replay based), N-LoRA and OLieRA to our results table. Our updated results show that SuRe maintains its strong lead in the Large Number of Tasks setting (see Table 2). Additionally, we are working on including either InfoRS, MIR, or another recent replay-based approach before the end of the discussion period.
>
> [c] Yujie Feng, Jian Li, Xiaoyu Dong, Pengfei Xu, Xiaohui Zhou, Yujia Zhang, Zexin LU, Yasha Wang, Alan Zhao, Xu Chu, and Xiao-Ming Wu. Aimmerging: Adaptive iterative model merging using training trajectories for language model continual learning, 2025.

---

### Official Review · Reviewer_Bq5a · 2025-11-02

**Soundness:** 4
**Presentation:** 4
**Contribution:** 3
**Rating:** 8
**Confidence:** 4

**Summary:**

The paper attributes catastrophic forgetting in continual LLM learning to replay selection and integration errors. It proposes to rectify these errors by selecting 'surprising' examples (characterized by high nll) to replay, and learning a 'slow' learner that is updated using EMA over the 'fast' learner (i.e., the one optimized directly over the incoming and buffered samples). The two learners are implemented using LoRA adapters. Empirical evaluations support the claims and provide appreciable improvements over baselines.

**Strengths:**

- The paper is well-written, claims are intuitive, and theoretically and empirically validated.
- To the best of my knowledge, the paper is the first to propose NLL for sample selection. Combining this with a slow learning strategy empirically shows significant improvements, as shown in Table 1.
- SuRE is implemented using LoRA and is therefore architecture agnostic.
- Evaluations and ablations are sufficient, and SuRE outperforms SOTA continual LLM learners on both benchmarks.

**Weaknesses:**

- To my knowledge, there are no significant weaknesses.

**Questions:**

- What is the value of $\beta$ used for evaluation? Can the authors include an ablation to show its impact?
- Can the reliance on high surprise cause the model to overfit to outliers (such as mislabeled samples) and inadvertently hurt the performance of the model? Perhaps buffering a combination of surprising samples and some randomly selected samples instead can help?

---

> ### Author Response · Authors · 2025-11-26
>
> We thank the reviewer for their positive assessment and for highlighting that our claims are theoretically and empirically validated. We are particularly encouraged that you found our NLL-based selection strategy novel and effective. We have addressed your specific questions regarding the hyperparameter $\beta$ and outlier sensitivity below.
>
> ### Question 1:
> We thank the reviewer for pointing out this omission. The value of $\beta$ used for training is 0.995. Following your suggestion, we also conducted an ablation study, which is reported in Table 10 of the updated version of the paper. Increasing $\beta$ to 0.999 resulted in a substantial drop in performance for all the replay-EMA variants tested, while decreasing the value from 0.995 to 0.99 and 0.985 led to a more gradual decrease in performance in all cases.
>
> ### Question 2:
> This is a very valid point, and it could indeed pose a risk in the case of label-level surprise. However, we believe this risk is already mitigated by using full-sequence surprise (and not label level surprise). In fact, this approach reduces the impact of extreme outliers while still focusing on “harder” samples. The issue you highlighted was likely minimal in the two datasets we evaluated (as they probably do not include any mislabelled datapoints). Nevertheless, in future studies, a buffer that combines surprising and random samples might be necessary when extreme values or outliers are more prominent.

---

### Author Response · Authors · 2025-11-26
**General Response: Updates on Baselines, Theory, and Ablations**

We thank the reviewers for their constructive feedback and rigorous assessment. We are pleased that reviewers found our problem decomposition "novel" (NcZB), our method "straightforward" (uMAn), our paper well-written (Bq5a) and our results "promising" (n5Un).

Based on the common themes in the reviews, we have updated the manuscript significantly. Below is a summary of the major changes:

**1. Expanded Baselines & The "Scale" Argument (Addressing uMAn, n5Un)**
We have added **Table 2** to include recent strong baselines: **AimMerging, N-LoRA, and OLieRA**.
* **Result:** While these baselines are highly effective on the short *Standard CL* benchmark (4 tasks), **SuRe + EMA** significantly outperforms them on the rigorous **Large Number of Tasks (LNT)** benchmark (15 tasks), achieving gains of up to **+5 points**.
* **Takeaway:** We argue that Standard CL is approaching saturation (near MTL performance). Our results demonstrate that SuRe is specifically designed for **scale**, effectively mitigating the integration error that accumulates over long task sequences.

**2. Clarifying Novelty: Theoretical Unification (Addressing uMAn, NcZB, n5Un)**
We explicitly acknowledge that we did not invent Dual-Learners or EMA. We have revised the introduction and related work to cite *DualNet* and *Slow-Fast* approaches as our architectural foundation.
* **Our Contribution:** We provide the **theoretical unification** of these components. We prove (Theorem 1) that catastrophic forgetting decomposes into two additive terms: *Selection Error* and *Integration Error*.
* **Complementarity:** We show that Replay (SuRe) and Dual-EMA are mathematically **complementary** solutions. Optimising one cannot fix the other. This theoretical insight explains *why* combining them yields the super-additive empirical gains observed in our experiments.

**3. Metric Clarification (Average vs. Sum)**
Prompted by **Reviewer uMAn**, we rigorously re-verified our implementation. We confirmed that our code uses **Average NLL** (normalised by length), not Sum. We have corrected **Equation 4** to reflect this.
* **Ablation:** We added a new ablation (Table 11 in the Appendix) comparing Average vs. Sum. Results confirm that **Average NLL** performs better ($\approx +1.0\%$ acc), as it prioritises *information density* rather than sequence length.

**4. Visual & Structural Updates**
* **Figure 1:** Revised to clearly distinguish between Gradient updates (Fast Learner) and EMA updates (Slow Learner).
* **Forgetting Metrics:** Highlighted BWT/FWT tables in the Appendix (Table 9) to provide a complete picture of stability.

We believe these revisions address the core concerns regarding novelty and comparative rigor, establishing SuRe as a strong baseline for scalable continual LLM fine-tuning

---

### Author Response · Authors · 2025-12-03
**Final Author Update**

We thank the reviewers for their time and thoughtful feedback throughout the discussion period. This engagement has truly helped improve and polish our manuscript.

As a final update, we are pleased to share the results of the additional experiments conducted following the specific recommendations of Reviewers **uMAn** and **n5Un**. These results have been included in the last version of the paper in Table 2, 4 and 5.

---

## 1. Expanded Baselines (Addressing uMAn & n5Un)

To ensure a comprehensive comparison against the state-of-the-art, we have added results for 4 new methods (Table 2), including two recent replay-based approaches:

- **AimMerging** (Replay-based)
- **Leitner Replay** (Replay-based)
- **N-LoRA**
- **O-LieRa**

**Result:**
Our method (**SuRe**) maintains state-of-the-art performance on the rigorous *Large Number of Tasks (LNT)* benchmark against these new baselines. This reinforces our claim that SuRe is uniquely capable of mitigating integration error over long task sequences, where other methods saturate.

---

## 2. Updated Ablation Studies (Addressing n5Un)

Per Reviewer n5Un’s request for performance analysis under budget constraints, we have updated our ablation studies regarding buffer size (Table 4) and replay ratio (Table 5).

**Result:**
SuRe remains robust and effective even in constrained settings (smaller replay buffers and lower replay ratios), consistently outperforming baselines.

---

## Conclusion

With the theoretical clarifications regarding the decomposition of forgetting (Theorem 1), the correction of the metric implementation (Average NLL), and now the inclusion of these requested baselines, we believe we have addressed all concerns raised during the review process, as already pointed out by Reviewer NcZB. We hope these final results provide the AC with a clear picture of the method's validity and performance.

---

### Meta-Review · Area_Chair_627R · 2025-12-29

**Summary:**

The leading strength in the paper is the theoretical decomposition, which several reviewers found novel and insightful, particularly the formal separation of forgetting into two additive terms (though the paper initially was not clear on how such terms mapped to architectural components).

The SuRe approach is also considered architecture-agnostic and practical, making it easy to integrate into existing LoRA-based continual fine-tuning pipelines.

The strong empirical performance is an additional strength of the work, although (relevant) questions were raised as concerns the inclusion criteria of baselines, which were missing several related replay-based methods.

On the side of weaknesses, the most relevant one pertains novelty. Both surprise-based prioritisation and slow–fast/EMA learners are known in literature, and the methodological contribution largely lies in their combination and analysis rather than inventing new mechanisms. Also, the Authors initially failed to appropriately reference such background in their work.

Reviewer uMAn initially raised doubts about the reliability of the surprise metric, including sensitivity to sequence length or outliers, which triggered Authors' clarifications whose arguments essentially hinge on the code being correct while the equation in the paper was wrong. This is a strong issue as code is not available with the submission.

Overall, the quality and correctness of the technical presentation was questioned by several reviewers (uMAn, NcZB, n5Un). The dual-learner mechanism was confusingly presented, and its benefit over a single learner required clearer theoretical and empirical justification.

On the side of the experimental analysis, the reviewers highlighted the lack of relevant related baselines from replay-based methods.

Several reviewers also noted that claims about “continual LLM learning” may be too broad, since experiments are limited to sequence-level text classification rather than CL in generative settings.

**Reviewer Concerns:**

The novelty of the dual fast–slow learner was clarified by explicitly acknowledging prior work and reframing the contribution as a theoretical unification, supported by Theorem 1 showing that selection error and integration error are additive and require complementary solutions. This clarifies the core of the contribution of the work, while it leaves open the point about the proposed architecture being an incremental advancement from a methodological perspective.

The issue about the confusing presentation of the method was, somehow, addressed by a revision of the central sections of the paper, including Figure 1. The overall impact of such a revision is hard to be determined as there is no visual tracking of the changes in the revised paper version.

Concerns about insufficient or outdated baselines were partially addressed by adding several recent replay-based and LoRA-based methods (e.g., AimMerging, N-LoRA, OLieRA). They show that SuRe remains the strongest on the LNT benchmark, but performance gains on the Standard CL benchmark are sometimes modest, leading to questions about generality beyond the LNT setting. Also, none of the performance metric has confidence intervals, which somehow reduces the trust on the results.

Post rebuttal, the evaluation remains limited to sequence-level text classification, leaving open whether the conclusions fully extend to generative continual learning for LLMs, despite softened claims in the revision.

**Reviewer Scores:**

Reviewer Bq5a provided an enthusiastic review of the paper with an high score (8) and medium-high confidence (4), associated to a very short an technically light review. The review itself highlighted no issues, while reports from other reviewers clarified that there were substantial technical errors in the formulations initially reported in the paper. I suspect that, if the reviewer had engaged during the rebuttal, they would have been forced to consider revising their score to a lower one (6?).

Reviewer uMAn was the most negative one (score 2), supported by concerns related with novelty, reference to literature and the robustness and adequacy of the surprise measure as described in the paper. The latter, in particular, did not receive a strong response by the Authors, which claim that the equation was indeed wrong but the implementation (which is not available to the reviewer) was correct. My understanding is that the reviewer would not have changed their mind during the rebuttal, and could have vouched against acceptance.

Reviewer NcZB had an initial weakly negative score (4), but in their response to the rebuttal they declared themselves satisfied with the answer. A score increase to 6 in this case was possible.

Reviewer n5Un score the paper as weakly negative (4). No response to rebuttal was given. The Authors' response was adequate concerning clarifying some technical aspects which were previously poorly presented. On the other hand, the issue about novelty remains, so I would expect the reviewer to remain with the current score after rebuttal.

Overall, the final average score of the paper post-rebuttal could be projected between 5 and 5.5. In consideration of the issues raised about the technical correctness of the surprise measure (and the lack of code in the supplementary), as well as of the limited novelty, I expect that this borderline paper would fall on the reject end.

---

### Decision · Program_Chairs · 2026-01-26

Reject